# Mechanism of RNA polymerase III termination-associated reinitiation-recycling conferred by the essential function of the N terminal-and-linker domain of the C11 subunit

Saurabh Mishra [1,2], Shaina H. Hasan[1,3], Rima M. Sakhawala [1,4], Shereen Chaudhry [1,5] & Richard J. Maraia [1✉]

RNA polymerase III achieves high level tRNA synthesis by termination-associated reinitiation-recycling that involves the essential C11 subunit and heterodimeric C37/53. The C11-CTD (C-terminal domain) promotes Pol III active center-intrinsic RNA 3′-cleavage although deciphering function for this activity has been complicated. We show that the isolated NTD (N-terminal domain) of C11 stimulates Pol III termination by C37/53 but not reinitiation-recycling which requires the NTD-linker (NTD-L). By an approach different from what led to current belief that RNA 3′-cleavage activity is essential, we show that NTD-L can provide the essential function of *Saccharomyces cerevisiae* C11 whereas classic point mutations that block cleavage, interfere with active site function and are toxic to growth. Biochemical and in vivo analysis including of the C11 invariant central linker led to a model for Pol III termination-associated reinitiation-recycling. The C11 NTD and CTD stimulate termination and RNA 3′-cleavage, respectively, whereas reinitiation-recycling activity unique to Pol III requires only the NTD-linker. RNA 3′-cleavage activity increases growth rate but is nonessential.

[1] Intramural Research Program of the Eunice Kennedy Shriver National Institute of Child Health and Human Development, National Institutes of Health, Bethesda, MD, USA. [2] Present address: Department of Biochemistry, Banaras Hindu University, Varanasi, Uttar Pradesh, India. [3] Present address: Mayo Clinic Alix School of Medicine, Scottsdale, AZ, USA. [4] Present address: Section on Regulatory RNA, National Institute of Diabetes and Digestive and Kidney Diseases, Bethesda, MD, USA. [5] Present address: Pfizer (Pearl River Site), 401 N Middletown Rd, Pearl River, NY, USA. ✉email: maraiar@mail.nih.gov

RNA synthesis occurs by evolutionarily related multisubunit RNA polymerases which in bacteria and archaea are comprised generally of 5 and 12 subunits, respectively. In eukaryotes, three homologous systems utilize Pols I, II, and III, composed of 14, 12, and 17 integral subunits[1]. Pol II transcribes the largest and most complex gene repertoire, responsible for all mRNAs and many noncoding (nc)RNAs, Pol I transcribes a single multicopy large rRNA gene, and Pol III is specialized to produce a great number of short transcripts which occurs by high-efficiency recycling on tRNA genes, and other ncRNA genes[1].

In yeast, Pol III produces tRNAs at ≥15-fold molar excess relative to ribosomes from ~275 dispersed tRNA genes[2]. Human pathogenic mutations are linked to subunits of Pol III transcription factors (TFs), and another class of diseases arises from dysregulation of ncRNA genes that are transcribed by Pol III[3]. Single-allele mutations in a small number of Pol III subunits cause a conditional phenotype, the severe outcome of infection by Varicella Zoster Virus (chickenpox), an otherwise self-limited disease, reflecting deficiency of an innate immune function that requires Pol III transcription activity in the cytoplasm[4–6]. More recently, it was reported that Pol III synthesizes the RNA of an RNA:DNA hybrid that serves to protect DNA 3′ ends during homologous recombination[7]. A large source of mutations that cause the syndromic disorder hypomyelinating leukodystrophy (HLD) occurs in the two largest Pol III subunits, whereas POLR3K/RPC10 the highly conserved small-subunit homolog of yeast C11 has also been a target[8–12]. Another complex disorder, Wiedemann–Rautenstrauch (aka neonatal progeroid) syndrome, and an expanding spectrum of pleiotropic phenotypes are also due to Pol III subunit mutations[13–19].

High transcript production by the Pol III system reflects the efficient transition from termination to reinitiation[20–22] attributable to its enzymatic properties as well as stability of the transcription initiation complex[23,24]. Much work including recent structural studies of the yeast components indicate that TFs IIIC and IIIB are required to assemble a stable complex on a tRNA gene prior to the first round of transcription by Pol III[25]. The two domains of IIIC, τB and τA, recognize the B- and A-box internal promoter elements[25]. The τA domain recruits TFIIIB (TBP, Brf1, and Bdp1) and helps it optimize binding upstream of the transcription start site (TSS) prior to Pol III recruitment[25]. Melting of the DNA strands occurs by allosteric action of Pol III-specific subunits assisted by Brf1 and Bdp1 resulting in bubble extension to the TSS[26–30]. Once assembled, TFIIIB remains fixed on the DNA and can direct recurrent reinitiations by Pol III[23,24].

Properties conducive of reinitiation–recycling are attributable to Pol III-specific subunits that function as intrinsic transcription initiation factors[1]. The trimeric and dimeric subunit complexes Rpc31/34/82 and Rpc37/53 are homologs of TFIIF and TFIIE, some of which facilitate Pol II preinitiation complex formation, and act in promoter opening[29–31]. However, in contrast to TFIIF and TFIIE which disassemble as Pol II elongates and must reassociate prior to subsequent initiation, C37/53 and C31/34/82 remain integrally associated with Pol III throughout the transcription cycle[29–31].

Pol III termination is also conducive to efficient recycling. First, it occurs by an autonomous and apparent simple mechanism in which RNA release is triggered upon reaching a short run of Ts in the nontemplate DNA[32,33]. Second, C37/53 increases the efficiency of termination and promotes reinitiation along with C11[22]. Thus in addition to the two largest subunits, Rpc2 and Rpc1 which comprise the active center, C37/53 and C11 increase Pol III termination and reinitiation–recycling[33–39]. C11 is a highly conserved protein of ~110 aa whose NTD is homologous to Pol II subunit Rpb9 and its CTD is homologous to the RNA 3′-cleavage domain of Pol II elongation factor TFIIS[40]. The NTD resides peripherally on Pol III, in contact with C37, the Rpc2 lobe, and Rpc1 jaw[41]. Recent structures of human Pol III (hPol III) confirm this and the overall general architecture of the 17 subunits in the corresponding elongation complexes (ECs)[42–45].

Although the C11 CTD was not visible in the yeast Pol III EC, it was resolved in the Pol III apoenzyme complex in which it was docked peripherally, although a connection path to the NTD was unresolved because the central linker region was not visible[41]. Cryo-EM structures of hPol III resolved its full-length C11-homolog, RPC10, in different conformational states of a transcription complex[43]. In one of these states, the NTD linker (L) turns inward and extends along the jaw into the funnel, projecting the CTD into the active center pore, positioning the tip of the acidic hairpin toward the elongating RNA 3′ end[43–45]. Conservation of sequence and structure by yeast C11 and human RPC10 is illustrated in Fig. 1a–c.

Likely facilitating termination is a loose RNA:DNA hybrid binding site that can readily release upon entering the first ≥5T tract encountered[41]. Initially documented for a yeast Pol III EC, the loose RNA:DNA-binding site was confirmed as a specific feature of hPol III EC[43]. Structures of hPol III ECs subjected to 3D variability analysis revealed that large movements of the C11 CTD and linker from their positions concomitant with elongation-termination was associated with the opening of the Rpc1 clamp (and presumed RNA release), suggesting that C11 may integrate activities involved in different phases of Pol III recycling[43].

The C31/34/82 initiation subunits associate with the Rpc1 clamp and dsDNA-binding site, whereas C11-NTD-C37/53 are at the jaw on the other side of this site[41]. A potential link connecting the RNA:DNA-binding site and the dsDNA-binding site was suggested by positive selection screens for loss-of-function (LOF) rpc11-alleles. Two different termination-related activities, read-through (R–T) and RNA 3′-cleavage, differentially mapped to its NTD and CTD[37,46]. Similar screens isolated termination R–T mutants in C37[38], as well as LOF and gain-of-function (GOF) mutants in Rpc1[47]. Evidence of connections between the Pol III periphery and active center was also obtained for C37/53 by biochemical and other approaches[28,48]. High-resolution structures have shown that parts of the C31/34/82 and C37/53 complexes are juxtaposed to downstream DNA-binding sites while other parts penetrate the active center[29,30,41]. Germane here is that the C11 NTD is docked at the periphery in contact with C37 while its CTD can reach the active center to rescue elongation-arrested Pol III via its RNA 3′-cleavage activity[40].

It is relevant to note that purified Pol III ECs terminate in a 5T terminator with ~50% efficiency, whereas the 14-subunit Pol III-core ECs lacking C11/37/53 terminate with ~10% efficiency[33,39,49]. Importantly, despite low termination efficiency at 5Ts, Pol III-core terminates with high efficiency (≥80%) in a 9T terminator, comparable to the 17-subunit holoenzyme although with more propensity to arrest than to read-through[33,49]. Studies indicate that an rU:dA hybrid of ≥8 bp spontaneously destabilizes the Pol III-core EC, presumably reflecting an innate trigger of the termination mechanism, and that C37/53/C11 can counteract the propensity to arrest[39]. A CTD region of C37 referred to as the initiation/termination loop[29,30,38,41,50] promotes termination to occur proximally in the T-tract which appears to work in conjunction with recognition of nontemplate (NT) DNA T-residues as a pause signal[33]. In addition, C37/53 appears to help maintain RNA 3′-end pairing with the template during termination, preventing arrest[39].

As was found for all Pol III subunits in *Saccharomyces cerevisiae*, the C11 subunit is essential for cell viability[22]. Multiple activities have been attributed to C11, and moreover, some data

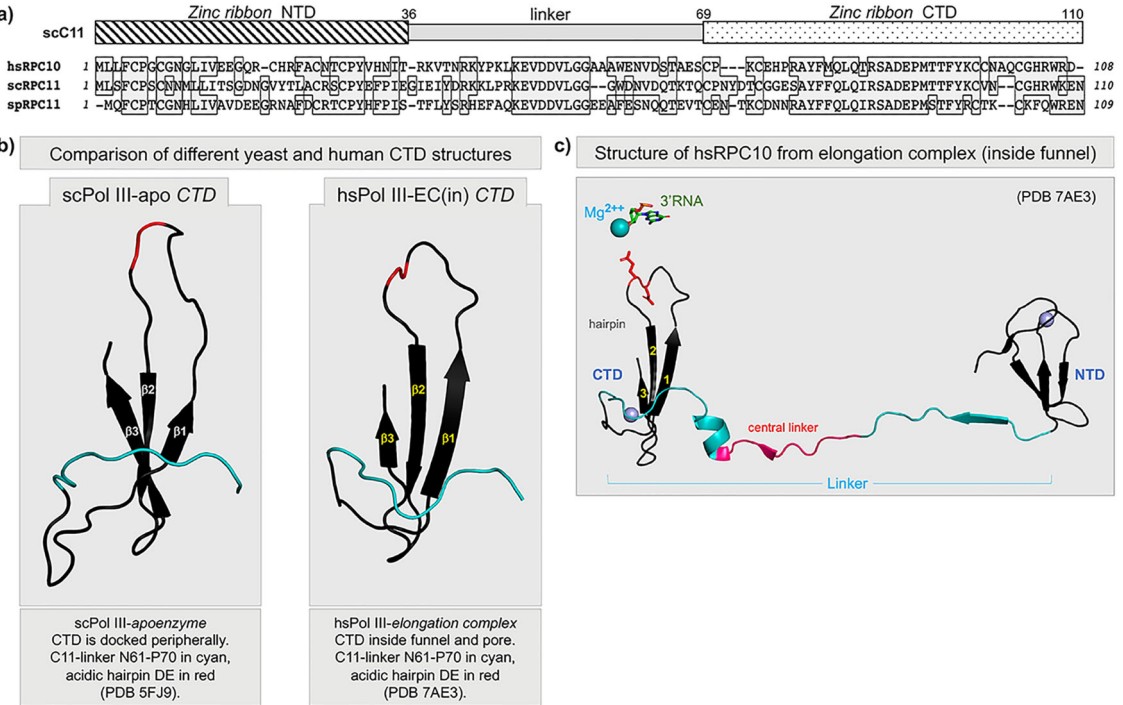

**Fig. 1 Sequence and structure conservation of RPC11 and RPC10 homologous subunits of yeast and human Pols III. a** Top: schematic of the three conserved domain structure consisting of an N-terminal domain (NTD) and a C-terminal domain (CTD) each comprised of a Cys-4 Zinc ribbon, connected by a linker domain. Bottom: alignment of the human (hs), *S. cerevisiae* (sc), and *S. pombe* (sp) sequences; identical amino acids are in shaded boxes. **b** Comparison of the C11 CTD taken from the *S. cerevisiae* Pol III apoenzyme (PDB 5FJ9)[41], and RPC10 CTD taken from the human Pol III elongation complex (inside the funnel conformation) (PDB 7AE3)[43]. The distal linkers are colored cyan and the positions of the acidic hairpin amino acids are in red. **c** Full-length RPC10 taken from the human Pol III elongation complex showing the NTD, extended linker, and CTD in the inside the funnel and pore position (PDB 7AE3). The acidic hairpin Glu-89 is shown in red stick format pointing toward an active site Mg++ (cyan sphere, at >7.5 Å) and the RNA 3′ nucleotide (stick format) which was noted to be too distant for interaction in this context[43]. The linker is comprised of a proximal part with a beta-strand colored cyan, an invariant central linker colored red, and a distal part with a short helix and extending beyond, in cyan. The two zinc molecules, one with the NTD and one with the CTD, are shown as gray spheres.

led to perplexing conclusions regarding its essential function. Prior to the discovery of the physical association of C11 with C37/53, the termination deficiency of purified Pol IIIΔ lacking the (mutant) C11 indicated its importance for termination[40]. It was then discovered that Pol IIIΔ deficiency was due to the lack of a newly uncovered subunit, C37 which unknowingly dissociated, with its heterodimeric partner C53, along with the mutant C11[22]. Thus the same study concluded that although C11 was not required for nor influenced Pol III termination, it was required for, along with C37/53, facilitated recycling by Pol III[22], the process of multiple cycle RNA syntheses from a stable transcription complex dependent on proper termination[20].

However, confounding the results were data that showed that C11 RNA-cleavage activity was not required for Pol III facilitated recycling[22] and that its cleavage activity was essential for viability[40]. Overexpression of cleavage-deficient mutant C11-D91A,E92A[40] (hereafter DE-AA) was toxic in *S. cerevisiae*, and this was confirmed with cleavage-deficient C11-(D90G) in *S. pombe*[46]. Further, C11 RNA-cleavage activity was deemed essential based on C11-(DE-AA) plasmid shuffle experiments when expressed under promoter-repressive conditions[40] (also C11-E92H see ref. [51]). Yet, errors due to lack of C11 cleavage-mediated transcription fidelity[51] are not expected to be consequential to short-length Pol III transcripts. Also, the RNA 3′-cleavage factor for Pol II TFIIS nor Pol I subunit Rpa12.2, are essential in *S. cerevisiae*[52–54]. It was unclear why C11 RNA-cleavage activity would be essential.

Prior studies showed that although not required for termination[22,37,39], C11 cleavage activity can occur at Pol III termination with functional outcome[46]. C11 was shown to stimulate the transcription termination by C37/C53[39]. Although the CTD was not visible in the yeast Pol III (yPol III) EC[41], the hPol III EC showed much of the linker and CTD in the funnel when the DNA clamp was open but was retracted to an outside the funnel position in a closed clamp structure[43]. Further analysis provided evidence for the important possibility that the mobile C11 linker and CTD participate in the dynamics of termination and recycling in addition to their role in providing RNA 3′-cleavage activity during transcription arrest and backtracking[43].

We examined *S. cerevisiae* C11 domains for various Pol III-dependent activities and for cell viability. The NTD alone stimulates termination by C37/53 but not reinitiation–recycling in vitro nor can it provide the essential C11 function. The NTD-L stimulates termination by C37/53 and is required for reinitiation–recycling, and contrary to current belief also provides the essential C11 function in vivo indicating vital importance of the linker. The data resolve misunderstandings about C11 function and advance insights into the mechanisms involved in termination-associated reinitiation–recycling by Pol III.

The collective data suggest that the NTD and CTD of C11 appear capable of independent function to stimulate termination and RNA 3′-cleavage, respectively, and when connected by the essential linker to confer the critical reinitiation–recycling activity that is unique to Pol III among the RNA polymerases.

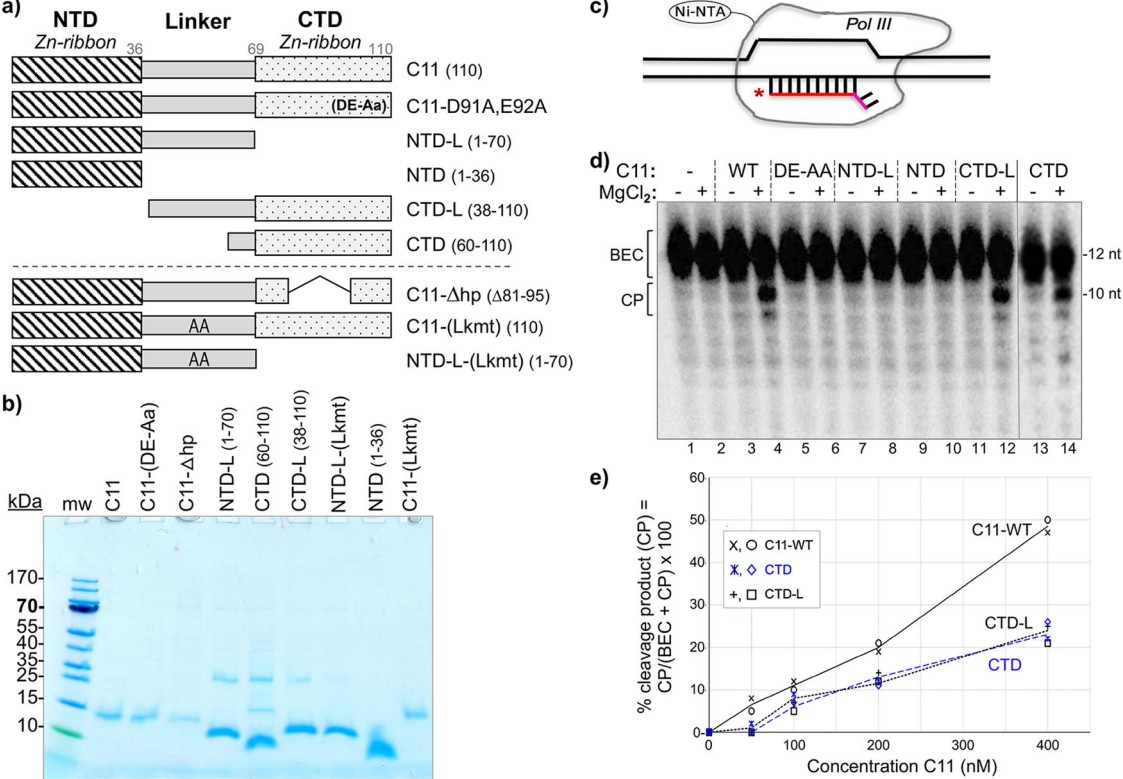

**Fig. 2 The isolated C-terminal domain (CTD) of C11 can activate RNA 3′-cleavage in backtrack arrested Pol III. a** Diagram of the C11 constructs used in this study; the ones above the dashed horizontal line are used in this figure. **b** Coomasie blue-stained gel of the C11 proteins used, corresponding to panel **a**. **c** Schematic of a DNA scaffold Pol III elongation complex (EC) in which the 3-end of the 12-nt RNA (depected in red) has a 2-nt mismatch with the template, representing an arrested EC with a RNA 3′-overhang in a backtracked state; the asterisk represents the radiolabeled 5′-$^{32}$P. **d** RNA 3′-cleavage assay results. A batch of backtracked arrested EC substrate was aliquoted to different reaction tubes and incubated with either buffer alone or recombinant C11 as indicated above the lanes, for 15 min prior to addition of MgCl$_2$ as indicated. Total RNA was analyzed. The starting material, backtracked EC, and cleavage products, BEC and CP, are indicated by brackets; the image in lanes 13–14 was from the same assay run on a different gel. Positions of 12-nt and 10-nt RNAs are indicated to the right. **e** The fraction of CPs, calculated as % cleavage product (CP) = CP/(BEC + CP) × 100, were quantified and calculated with Multigauge V3.2 software[39] and plotted against the concentration of C11, CTD, and CTD-L. Two data points for each concentration are representative of independent biological replicates for the same batch of C11 construct proteins, as indicated in the inset. The means of the two points were used to plot the lines for the proteins annotated. Additional assay data are shown in Supplementary Fig. 1a. Linear regression analysis of these quantified data with $R^2$ values is shown in Supplementary Fig. 1b.

## Results

### The isolated C11 CTD stimulates Pol III intrinsic RNA 3′-cleavage activity.

The NTD and CTD of C11 are comprised of Cys-4 zinc ribbon motifs connected by a conserved linker domain (Fig. 1). Figure 2a shows a schematic of the C11 constructs used for this or other figures, and the recombinant proteins purified, as described previously[39], are shown in Fig. 2b. The schematized substrate used in the assay to examine the protein constructs for RNA 3′-cleavage activity is shown in Fig. 2c[39,49]. Epitope-purified Pol III-core assembled as an RNA:DNA hybrid complex whose 5′-$^{32}$P-RNA has two mismatched nucleotides at the 3′-end mimics of a backtracked elongation complex (BEC), is a C11 substrate[39,49]. Prior analyses had shown that C37/53 tempers the backtracking activity of C11 in these BEC assays[39]. Our goal was to determine which of the C11 constructs could exhibit basic cleavage activity, somewhat similar to work in which purified Pol II ECs on tailed templates were supplemented with recombinant domains of TFIIS[55]. Therefore, C37/53 was not included in Fig. 2d which demonstrated active site cleavage by C11, CTD-L, and CTD by producing cleavage products (CP) in the presence of MgCl$_2$, whereas C11-(DE-AA), NTD, and NTD-L were inactive.

We compared CTD, CTD-L, and C11 (Fig. 2e) over a range of concentrations. Supplementary Fig. 1b shows the quantified data subjected to linear regression analysis. The observed concentration-dependence may reflect a higher affinity of full-length C11 for Pol III-core EC despite the absence of C37/53, consistent with NTD binding sites on yPol III that contributes to a multifaceted interface with Rpc2 and Rpc1[41]. This consists of contacts between the N-terminal part of the NTD and Rpc2 (helix aa positions 280–295) as well as hydrophobic packing and other type interactions at the jaw that involve Rpc1 positions 1177–1180 and 1256–1258 (PDB 5FJ8)[41]. We also note that the linker had no apparent influence on CTD in this assay (Fig. 2e) (also see Supplementary Fig. 1).

### C11 NTD is sufficient to stimulate C37/53-dependent termination by Pol III.

A genetic screen identified mutations in *S. pombe* C11 NTD positions Y30, F32, I34 (Y31, F33, I35 of *S. cerevisiae*) that decreased termination, manifested as R–T past a 6T terminator[37]. Assays using Pol III-core ECs showed that C37/53-dependent termination is stimulated by wild-type C11 or cleavage-defective C11-(DE-AA)[39,49]. Although initially observed on a 9T terminator (Supplementary Figure in ref. [49]), characterization of a 5T terminator led to a more quantitative assay[39] which we used here (Fig. 3a, b). Pol III termination in this system corresponds to RNAs with 3–5 uridines at the 3′ ends, annotated TZ to the left of Fig. 3[39]. Results were quantified, calculated for termination efficiency (TE), and plotted in Fig. 3c. Consistent with previous data,

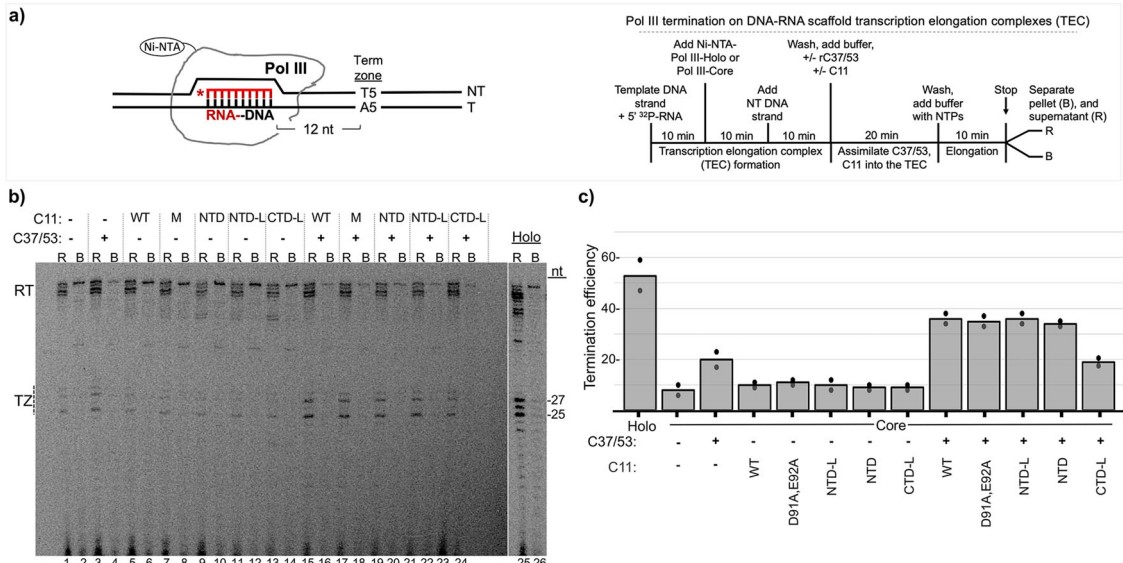

**Fig. 3 The isolated C11 NTD stimulates C37/53-mediated Pol III termination from scaffold ECs. a** Left: schematic of Nickel-Nitrilotriacetic acid (Ni-NTA) immobilized Pol III-core elongation complex (EC) assembled on DNA scaffold. The first T:A bp of the T5 terminator, indicated as Term Zone, begins after 12 bp downstream of the 3′ end of the 10-nt RNA primer (depicted in red); an asterisk indicates the 5′-$^{32}$P. The nontemplate (NT) and template (T) DNA strands are indicated. Right: schematic description of the Pol III transcription-elongation-complex (TEC) assembly and termination reaction. **b** Gel autoradiogram showing results of 10-min transcription-elongation–termination assay in the presence and absence of C37/53 and the C11 protein constructs indicated above the lanes, M = D91A, E92A (DE-AA) mutant. Transcripts that were released from or remained bound to the immobilized Pol III are indicated above the lanes as R and B, respectively. TZ denotes the positions of the termination products released in the Term Zone, and RT denotes read-through to the end of the DNA. Lanes 25 and 26 were from the same gel, same exposure. The −27 and −25 reflect RNA size markers in nucleotides (nt), these represent transcripts with 3 and 5 uridylates, respectively. **c** Graphic representation of quantification of transcription efficiency, quantified and calculated with Multigauge V3.2 software[39]. The bars indicate the means of two data points (black circles) for each condition which represent independent duplicate biological replicates for the same batch of the construct proteins as indicated below.

Pol III-holo was more efficient for 5T termination in the scaffold assay than Pol III-core+C37/53 + C11 using comparable units of activity ("Methods")[39]. Pol III-core exhibits low TE, ~10% which increases to ~20% by addition of C37/53 (Fig. 3b, lanes 1–2 vs. 3–4, Fig. 3c). The addition of C11 alone has negligible effect, whereas either C11(WT) or C11-(DE-AA) substantially increase C37/53-dependent TE by Pol III-core, reproducing previous data[39]. Strikingly, NTD and NTD-L stimulate C37/53-dependent termination as much as C11 (Fig. 3b, c). By contrast, CTD-L which is active for RNA 3′-cleavage (Fig. 2c, d) fails to stimulate C37/53-dependent termination (Fig. 3b, c).

Figures 2 and 3 demonstrate that Pol III intrinsic RNA 3′-cleavage activity and autonomous termination activity can be distinctly stimulated by the isolated C-terminal and N-terminal domains of its C11 subunit, respectively, each without the linker. The RNA 3′-cleavage-stimulatory activity requires the acidic hairpin (residues DE) whereas termination-stimulatory activity does not. The data are consistent with genetic screens that yielded LOF C11 mutants in the RNA 3′-cleavage and termination activities of Pol III[37,46].

**C11 is required for C37/53-activated reinitiation by Pol III, despite efficient termination**. To examine which C11 domains are required for reinitiation–recycling, we isolated immobilized transcription initiation complexes that could be reconstituted with Pol III and recombinant C37/53 and C11. Supplementary Fig. 2a shows a schematic preparation of the transcription complexes. Supplementary Fig. 2b shows transcription products obtained from complexes at three stages of the protocol. Specifically, after formation and wash, a short period of transcription with unlabeled NTPs and +/− heparin is done to examine the effects of stripping away the native Pol III, followed by another wash. Synthesis of newly

transcribed T1 RNA in the presence of $^{32}$P-αNTP from the +heparin complexes is then dependent on the addition of purified Pol III-holo (Supplementary Fig. 2b, lanes 1–3). Supplementary Fig. 2c shows a Pol III-holo batch reaction equivalent to lane 3 in Supplementary Fig. 2b from which equal aliquots were removed and processed at the times indicated, providing evidence for recycling.

A batch of immobilized SUP4-tRNA$^{Tyr}$ gene preinitiation complexes was formed for 20 min according to the schematic on the right side of Fig. 4a, washed and aliquoted to five reaction tubes. To this was added Pol III-holo or Pol III-core + /−C11, + /−C37/53 as indicated. After Pol III initiates RNA synthesis, the T1 transcript indicates termination in the proximal 7T tract of the 7T-G-6T terminator (Fig. 4a), whereas T2 indicates read-through of 7T into the distal 6T tract. Indeed, the SUP4-tRNA$^{Tyr}$ terminator has been used to examine TE effects of C37/53[40]. The transcription products are shown in the gel in Fig. 4b.

A time course of this type for Pol III-core +C11, + C37/53, +C37/53 + C11, and Pol III-holo has not been described. Pol III-holo (Fig. 4b, lanes 1–5) and Pol III-core+C37/53 + C11 (lanes 21–25) appeared to make efficient use of the PICs, whereas Pol III-core alone or with +C11 alone or +C37/53 alone produced less RNA (lanes 6–10, 11–15, 16–20). The data were quantified (Fig. 4c) and subjected to regression analysis (Supplementary Fig. 3). The reproducible data points for each of the three duplicate transcription reactions containing Pol III-core alone or reconstituted with +C37/53 or +C11, reflected time-course curves that differed from those produced by reactions with Pol III-core+C37/53 + C11 and Pol III-holo (Fig. 4c).

The earliest samples examined were at 2 min for all of the reactions. Extrapolation to time 0 suggests immediate efficient engagement by Pol III-core+C37/53 + C11 and Pol III-holo, and recycling thereafter (Fig. 4c). However, those two reactions

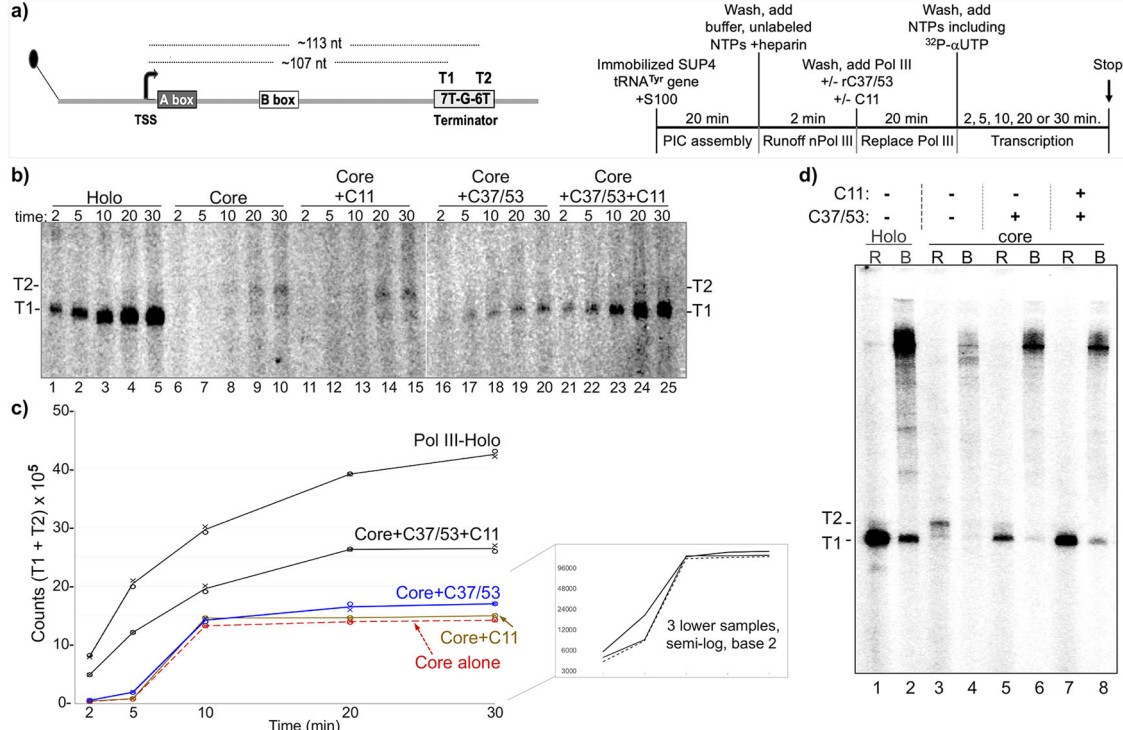

**Fig. 4 C11 activates C37/53 Pol III transcription factor (TF) complexes with efficient termination for reinitiation. a** Left: schematic representation of the immobilized SUP4-tRNA^Tyr gene with A-box, B-box promoter elements, and T1 and T2 terminators indicated, that is used to form transcription preinitiation complexes (PICs) described in the text and in Supplementary Fig. 2. Right: schematic description of the PIC assembly and transcription time-course reactions shown in (**b**); NTPs nucleoside triphosphates. **b** RNA products of in vitro transcription reactions carried out on a batch of SUP4-tRNA^Tyr transcription complexes reconstituted with either Pol III-holo (lanes 1–5, numbered below), Pol III-core (lanes 6–10), or Pol III-core preincubated with C11 only (core + C11), C37/53 only, or C37/53 + C11 subunits (lanes 11–15, 16–20, and 21–25). All lanes were part of the same transcription experiment and were run on the same gel. Lanes 15 and 16 were separated by empty lanes but were juxtaposed in the image shown as indicated by the vertical line. In this gel system, the T1 and T2 bands correspond to sizes ~107 and ~113 nucleotides (nt), respectively, as illustrated in panel **a** and annotated on the left side of panel **b** (see Supplementary Fig. 2b). **c** The amounts of T1 + T2 RNAs at each time point in **b** were quantified and calculated with Multigauge V3.2 software[39]. Two data points for each time point represent independent duplicate biological replicates for the same batch of construct proteins. The means of the two points were used to plot the lines for the sets of proteins as annotated. The blue and red lines were used for visual contrast. The inset shows only the lower three time-course plots on a semi-log scale, base 2. Regression analysis of the quantified data with $R^2$ values is shown in Supplementary Fig. 3. **d** Release of RNA at the T1 terminator is not sufficient for efficient recycling by Pol III-core. The RNA products of an in vitro transcription reaction as in panel **b** (30 min.) were separated into released (R) and bound (B), prior to examination by denaturing gel electrophoresis; the T1 and T2 bands are as in panels **a** and **b**, corresponding to ~107 and ~113 nucleotides (nt) in length.

produced time-course curves that were in contrast to and revealed a lag in the appearance of RNA products in the reactions for Pol III-core, Pol III-core+C37/53, and Pol III-core +C11 (Fig. 4c). The data indicate that Pol III-core is inefficient for reinitiation on preformed initiation complexes as is Pol III-core reconstituted with either +C37/53 or +C11 whereas Pol III-core reconstituted with both +C37/53 and +C11 is distinctly more efficient.

Supplementary Fig. 2 indicates that the initiation complexes had contained native Pol III as expected because ^32P-αUTP incorporation produced T1 RNA. As per the method of complex preparation, the native Pol III was run off the template, captured by heparin, and washed away, prior to use for Fig. 4b. Therefore, the predominance of T2 RNA in lanes 9, 10, 14, 15 of Fig. 4b suggests that Pol III-core initiated transcription from SUP4-tRNA^Tyr TFIIIB complexes that were previously transcribed by a different Pol III. Further, that T2 is predominant in lanes 6–15 suggests that the 7T-G-6T terminator of SUP4-tRNA^Tyr transcription complexes is recognized similarly as is a synthetic 9T terminator of scaffold ECs with Pol III-core. This suggests that T1 and T2 reflect Pol III-holo and Pol III-core mechanisms of termination, respectively[33,39,49].

Another set of reactions were separated into released and bound fractions (Fig. 4d). The majority of T1 RNA from Pol III-holo was released (lanes 1, 2) as it was for Pol III-core+C37/53 + C11 (lanes 7, 8) and Pol III-core+C37/53 (lanes 5, 6). Significantly, the relatively small amount of T1 RNA and a larger amount of T2 RNA synthesized by Pol III-core from the complexes were released (lanes 3, 4). The data in Fig. 4 corroborate the report using maiden SUP4-tRNA^Tyr complexes assembled with TFIIIB/TFIIIC for reconstitution with C37/53 and C11, analyzed for RNA release by a different approach[22]. Figure 4 shows that despite efficient Pol III-core termination at T1 or T2, and whether termination at T1 is activated by C37/53 or not, Pol III-core is deficient for reinitiation–recycling unless C11 and C37/53 are present together.

That more recycling activity was apparent with Pol III-holo than with Pol III-core+C11 + C37/53 was reproducible and is noteworthy. Specific aspects of this are addressed in "Discussion".

**C11 NTD with linker (NTD-L) is sufficient to activate C37/53-dependent Pol III recycling.** Cleavage-inactive C11-E92H had been shown to support Pol III recycling[22]. In agreement with this,

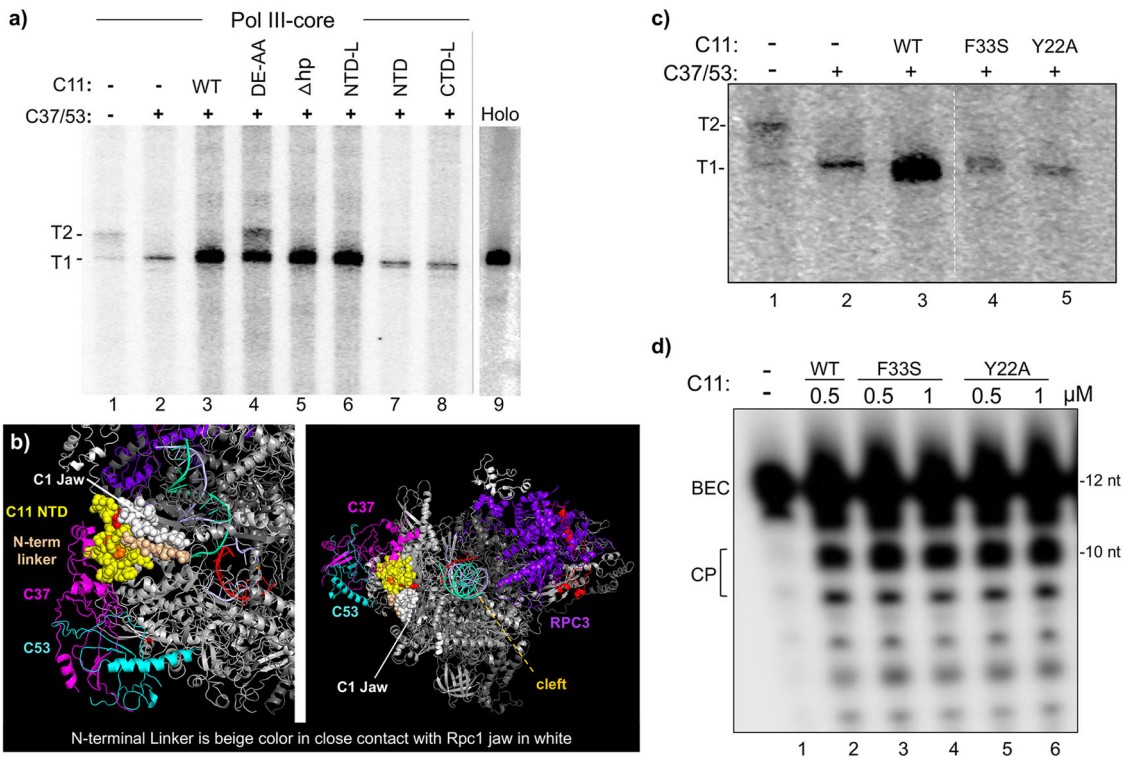

**Fig. 5 The isolated NTD-L of C11 can activate C37/53 for Pol III reinitiation–recycling. a** In vitro transcription reactions using a batch of SUP4-tRNA$^{Tyr}$ transcription initiation complexes reconstituted with either Pol III-holo (lane 9), Pol III-core alone (lane 1), or Pol III-core preincubated with recombinant C37/C53 alone, or in combination with different derivatives of C11 as indicated; the T1 and T2 bands are as described for Fig. 4a, b, d. Here, we also used cleavage-defective C11-Δhp, a β-hairpin deletion of amino acids 81 to 95 (Fig. 2a); lane 9 was part of the same transcription reaction experiment and was juxtaposed to the rest of the gel for reference. **b** Different views of cryo-EM structure model of *S. cerevisiae* Pol III EC highlighting key mutations examined in C (PDB 5FJ8)[41]. C37 is magenta and C53 is light blue, represented as ribbons. The C11 NTD is represented by yellow spheres. The N-terminal linker region is shown as tan spheres as it extends along the jaw region of Rpc1 represented as white spheres, toward the funnel. Phe-33 (F33) is orange and Tyr-22 (Y22) is red. **c** Each of the single-point mutations, F33A and Y22A to the C11 NTD, impairs reinitiation–recycling. Lanes 4 and 5 were part of the same reaction and gel as lanes 1–3 and were juxtaposed in the image shown as indicated by the dashed vertical line. **d** The F33A and Y22A mutations that impair C11-activated reinitiation–recycling do not impair CTD-mediated RNA 3′-cleavage activity.

C11-(DE-AA) was active for Pol III recycling (Fig. 5a, lane 4). More striking was that the CTD hairpin and the entire CTD were dispensable for this activity, as observed for the C11-Δhp (deleted hairpin) and NTD-L constructs, but not the linker which was non-dispensable (Fig. 5a, lanes 5–7). Figure 5a lanes 1 and 2 show low-level Pol III-core activity without C11 whereas lanes 3–6 show robust activity for C11, C11-(DE-AA), C11-Δhp, and NTD-L. C11-Δhp is deleted of most of β1 and the hairpin loop through β2 of scC11 (Figs. 1b and 2a). It was repeatedly observed that C11-(DE-AA) led to the production of T2 RNA whereas other RNA 3′-cleavage-inactive C11 proteins did not (Fig. 5a, lane 4 vs. 3, 5–6, 7–8). This is addressed in the next section.

We examined point mutations to the NTD. As noted above, C11-F33 is homologous to *S. pombe* C11-F32, identified together with Y30 and I34 in a LOF termination R–T screen (overexpression in wild-type background)[37]. Y31, F33, and I35 are in a region that interacts with a C37 extension (aa 130–140) (PDB 5FJ8)[41]. In addition, Tyr-22 interacts and fits between Rpc1 residues 1180 and 1275 (PDB 5FJ8). Thus C11 Y22 and F33 help anchor the C37/53/C11 trimeric complex at the jaw (Fig. 5b). Y22A and F33A each severely impaired C11 for Pol III recycling but not for RNA 3′-cleavage activity (Fig. 5c, d).

**C11 acidic hairpin mutations DE-AA inhibit C37/53-dependent termination.** A fraction of Pol III-core+C37/53 + C11-(DE-AA) failed termination at T1 and extended to T2 (Fig. 5a, lane 4). Figure 6a shows that this occurs in a concentration-

dependent manner and is specific to C11-(DE-AA); not observed for C11-Δhp nor NTD-L (Figs. 5a and 6a and Supplementary Fig. 4). Note also that C11-(DE-AA) stimulated rather than inhibited C37/53-dependent termination by Pol III-core in single-round factor-independent reactions (Fig. 3) consistent with the previous results[39].

**Cleavage-defective C11 CTD-containing mutants interfere with termination in vivo.** Heterologous yeast and suppressor-tRNA reporter systems have been used to study C11 function[40]. For this work, we developed a suppressor-tRNA allele to be dependent on RNA release in a short T tract[33,49] in *S. pombe* (T1 in Fig. 6b). This allele is high activity suppressor-tRNA with a 4T terminator followed by a cRT region that prevents productive processing of transcripts that read-through beyond T1[56]. Strain ySHA24 contains the allele depicted in Fig. 6b which exhibits a basal level of tRNA-mediated suppression (TMS) of *ade6-704* as has been characterized for a number of other suppressor-tRNA alleles[57–61].

Insufficient termination by Pol III in this system leads to the accumulation of red pigment. Validation was by testing previously characterized Rpc1 subunits[34]. *S. pombe* Rpc1-E850K is a LOF mutation in the bridge helix, whereas Rpc1-F1069L is a GOF mutation in the trigger loop which produced darker and lighter color respectively relative to empty vector, pRep4X, and Rpc1 (Fig. 6c, upper left) (correspond to E870 and F1111 in *S. cerevisiae* Rpc1[34]. Cleavage-deficient C11-D90G,R107C and

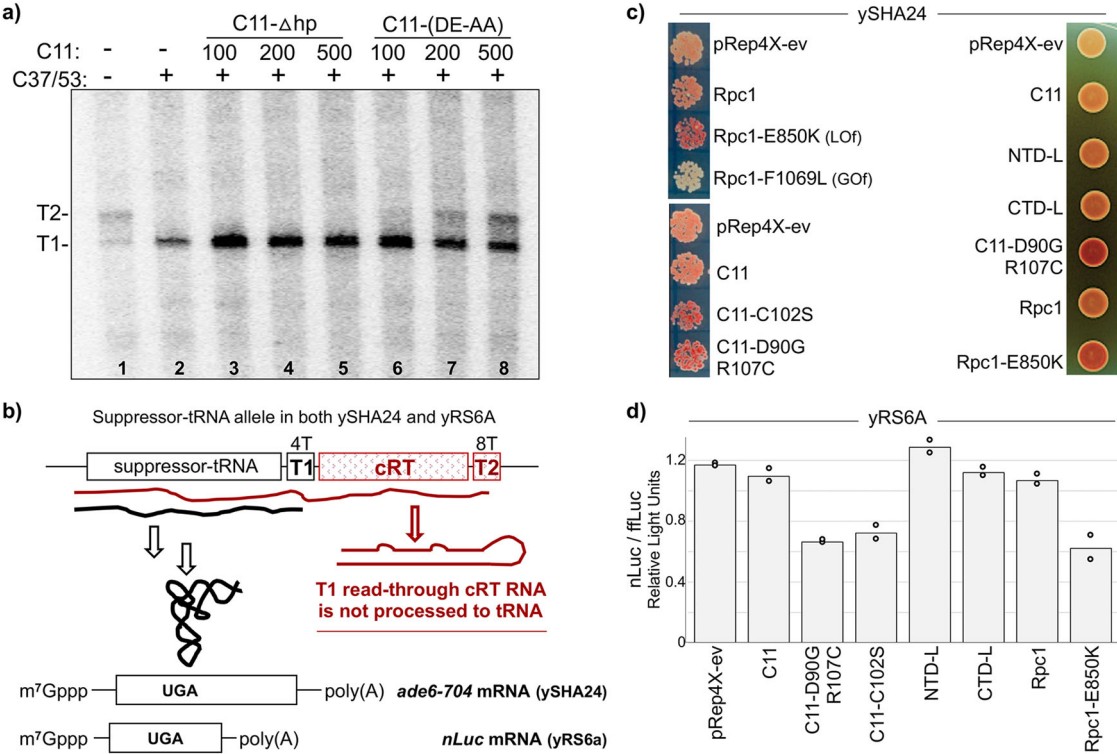

**Fig. 6 C11 cleavage-defective CTD mutants interfere with termination during Pol III recycling. a** In vitro transcription recycling reactions using the SUP4-tRNA[Tyr] transcription complexes reconstituted with Pol III-core (lane 1), or aliquots of a batch of Pol III-core preincubated with C37/53 alone (lane 2) or plus increasing amounts of C11-Δhp (deleted hairpin) or C11-(DE-AA) at 100, 200, and 500 nM, as indicated. The T1 and T2 bands are as described for Fig. 4a, b, d. **b** Schematic representation of the suppressor-tRNA gene in *S. pombe* strains ySHA24 and yRS6A used in **c** and **d**. The complementary read-through (cRT) region can form extensive base pairing with the tRNA region, therefore if it is transcribed will interfere with the formation of a functional tRNA, making suppression dependent on termination at T1[56]. **c** tRNA-mediated suppression (TMS) phenotypes in the red–white *ade6-704* assay as described in the text. Lighter color indicates increased termination at T1, a darker color indicates decreased termination at T1. **d** Graphic representation of quantification of relative activities in the dual nanoluciferase/firefly-luciferase TMS assay described in the text; nLuc and fflyLuc relative light units (RLU) are arbitrary. Bars indicate the means of two data points (circles) for the test constructs indicated representing independent biological replicates.

C11-C102S[46] led to decreased TMS (Fig. 6c, lower left). Cleavage-inactive NTD-L showed no positive or negative activity as compared to C11-D90G,R107C or Rpc1-E850K (Fig. 6c, right), consistent with the biochemical data that it did not interfere with termination. CTD-L showed no positive or negative activity (Fig. 6c).

Another TMS assay using the same suppressor-tRNA allele but in *S. pombe* strain yRS6A which carries a nanoluciferase (nLuc) suppressible reporter gene along with firefly-luciferase (ffLuc) gene as a separate transcription unit as an internal control. The nLuc/ffLuc TMS assay data (Fig. 6d) confirmed that C11-D90G,R107C and C11-C102S were inhibitory to suppression similar to Rpc1-E850K, consistent with interference with Pol III termination (Fig. 6d).

**C11 can activate C37 CTD-termination loop mutants for recycling despite termination deficiency.** C37/53 has been shown to function in termination, Pol III reinitiation–recycling[22,33,38,40], and promoter opening[28]. A screen for LOF mutants in *S. pombe* C37 identified a mutation hotspot corresponding to residues 226–230 of the *S. cerevisiae* protein[38]. Involvement of this region was supported by sequence-specific effects of the nontemplate (NT) DNA strand on pre-termination pausing by Pol III in promoter-independent single-round assays[33]. Although this region was not visible in Pol III EC structures it could be modeled next to the melted NT DNA, referred to as the termination/initiation loop (Fig. 4a in ref. [41]). This region was resolved in Pol III initiation

complexes associated with elements from TFIIIB Bdp1 and C34 during promoter opening[29,30].

Heterodimers with C37 position 226–230 mutations were examined for activity on the SUP4-tRNA[Tyr] transcription initiation complexes; C37*/53 in which the five amino acids are substituted with alanine, and C37[D]/53 in which they are deleted. These mutations had been shown to diminish termination in a single-round, promoter-independent assays, although were not compared to each other[33,39]. Lanes 1–4 of Fig. 7a show the heterodimers in nonrecycling conditions, in the absence of C11, while lanes 5–10 show activity on the same complexes with C11. In the absence of C11, the termination deficiency of Pol III-core, reflected by T2 RNA is corrected by WT C37 (lanes 1–2). By contrast termination deficiency with C37[D] was comparable to Pol III-core alone (lanes 1, 4), and also significant for C37*. The C37[D] mutation would appear to render C37 nonfunctional for Pol III termination on nonrecycling transcription complexes.

Lanes 5–7 of Fig. 7a show that C37*/53 and C37[D]/53 are active for C11-dependent Pol III-core reinitiation–recycling on the transcription complexes, although not without notable effects. The mutated C37 reactions produced less RNA than WT C37, and more so for C37[D]. The termination deficiencies reflected by the ratio of T2:T1 in the nonrecycling reactions were apparently suppressed with recycling in the presence of C11 (compare lanes 3–4 and 6–7). The cleavage-deficient C11 NTD-L also activated the C37* and C37[D] proteins for Pol III recycling (Fig. 7a, lanes 8–10). Again, reduction in the amount of T1 RNA was observed with C37* and C37[D] relative to WT C37 although the amount of

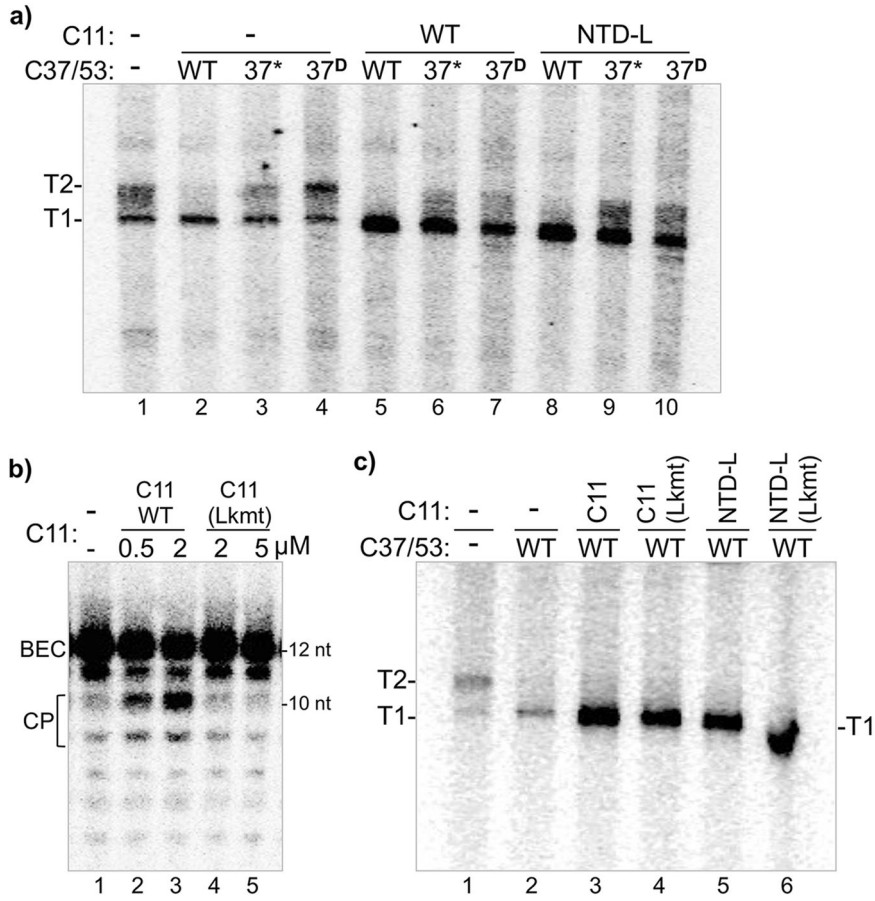

**Fig. 7 C11 can activate C37 CTD-termination loop mutants for recycling. a** The NTD-L of C11 can activate C37-termination/initiation loop mutants for Pol III recycling. In vitro transcription reactions using a batch of SUP4-tRNA$^{Tyr}$ complexes reconstituted with Pol III-core alone (lane 1), or Pol III-core preincubated with C37/53 or C37*/53 in which the five amino acids at positions 226–230 are substituted with alanine, or C37$^{D}$/53 in which they are deleted, alone (lanes 2–4), with C11 (WT, lanes 5–7) or with the NTD-L (lanes 8–10), as indicated. The T1 and T2 bands are as described for Fig. 4b and d. **b, c** Invariant linker mutations affect RNA 3′-cleavage activity but not Pol III recycling. The two C11 constructs with D52A,D53A mutations to the invariant central linker (Lkmt) are shown in Fig. 2a. **b** The RNA 3′-overhang substrate representing Pol III backtrack arrested ECs was incubated with either buffer alone, C11 or C11-(Lkmt) in amounts indicated above the lanes under standard assay conditions. Cleavage products (CP) are indicated. **c** Results of in vitro transcription reactions using a batch of SUP4-tRNA$^{Tyr}$ complexes reconstituted with Pol III-core alone (lane 1) or aliquots of a batch of Pol III-core preincubated with C37/53 together with C11, NTD-L, or the linker-mutated versions thereof as indicated above the lanes (the misshapen band in lane 6 resulted from a similarly misshaped loading well in the gel). The T1 and T2 bands are as described for Fig. 4b and d.

T2 RNA and intermediates between T1 and T2 was observed with the cleavage-deficient NTD-L appeared greater than with WT C11.

**The central invariant C11 linker is important for RNA 3′-cleavage activity.** The ~31 aa linker domain of C11 has been conserved in eukaryotes; the central nonameric sequence, KEVDDVLGG is identical in *S. cerevisiae*, *S. pombe*, and human (Fig. 1a)[40]. We mutated the two invariant aspartates to alanine, creating C11-(D52AD53A) designated C11-(Lkmt) (Fig. 2a) and examined it for RNA 3′-cleavage activity which revealed it to be defective including at higher concentration than WT C11 (Fig. 7b). Recent Pol III structures suggest that these mutations would disrupt contacts with highly conserved basic residues in Rpc1 that would presumably be important to guide the linker and CTD through a very narrow groove with invariant contacts to Rpc1, to the active center[44].

We also examined the C11-(Lkmt), NTD-L, and the corresponding NTD-L-(Lkmt) with the same mutations for the ability to activate C37/53-dependent recycling by Pol III-core (Fig. 7c). In both C11 forms, the D52A–D53A linker mutations reproducibly supported activation of C37/53 for Pol III

reinitiation–recycling, with no evidence of effects on termination (Fig. 7c).

**The isolated NTD linker confers the essential function of C11 to S. cerevisiae.** Either of two cleavage-defective alleles, C11-(DE-AA) and C11-E92H tested did not support viability in the absence of *RPC11*[40,51]. As noted, C11-(DE-AA) is toxic when over-expressed, as well as C11-(D90G) in *S. pombe*[40,46]. Our data above suggest that C11-(DE-AA) is not merely cleavage-deficient but that by its ability to insert into the active center it interferes with C37/53 function. This prompted re-examination of the essentiality of the C11 CTD and its stimulatory Pol III intrinsic RNA 3′-cleavage activity.

We employed the plasmid shuffle technique[62] to ask which if any *S. cerevisiae* (sc)C11 constructs could support viability in the *rpc11Δ* heterocomplemented strain yGAKL which has been a source of Pol III-core (aka pol IIIΔ) used for in vitro assays[28,33,39,49,60]. Deletion of the essential *RPC11* gene in *S. cerevisiae* yGAKL is complemented by a plasmid expressing *S. pombe* (sp)C11 from a *GAL1*-inducible promoter on a *URA3* plasmid[28]. By the exchange strategy depicted in Fig. 8a, the spC11 *URA3* plasmid is counter-selected against by 5-fluoroorotic acid

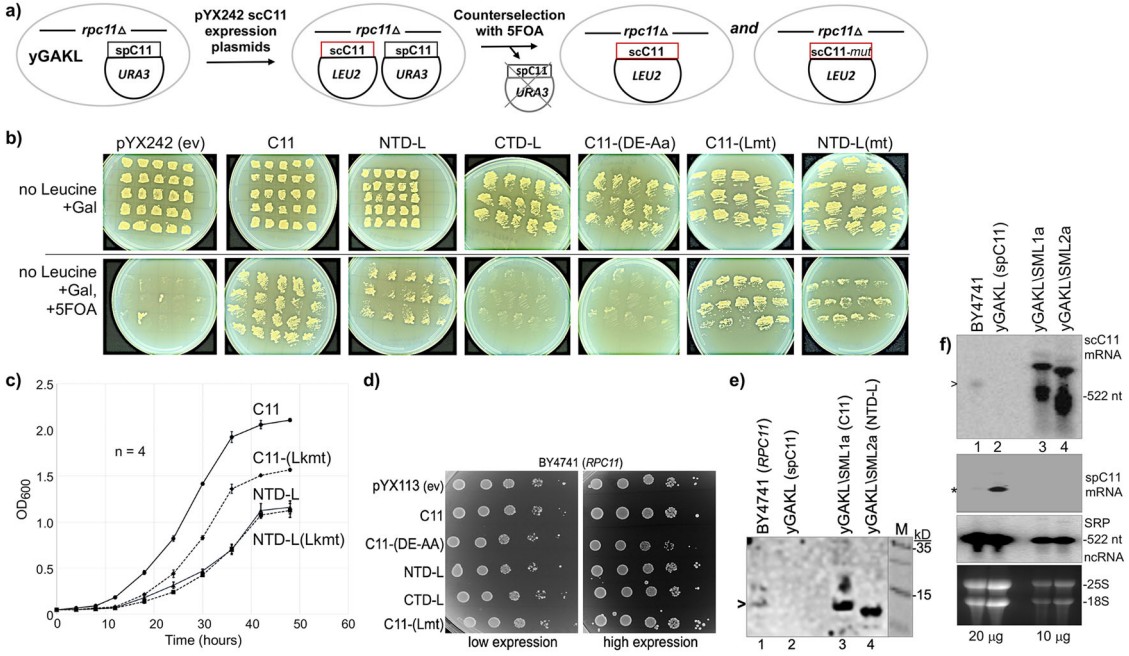

**Fig. 8 The NTD-L is sufficient to confer the essential function of C11 in _S. cerevisiae_. a** Schematic of plasmid shuffling experiment. The haploid strain yGAKL, _rpc11_Δ::Kan[R] complemented with a pRS316-_S. pombe_-C11 expression plasmid was transformed with pYX242 expression plasmids containing no insert (ev) or the six different constructs of _S. cerevisiae_ (sc)C11 indicated above the panels in **b**. Transformants were grown, and counterselection was then applied against the spC11 _URA3_ plasmid in media containing 5-fluoroorotic acid (5FOA) expecting to obtain scC11 (WT) and any mutant construct (mut) that might confer viability. **b** Initial individual transformants obtained after steaking in patches to plates containing media lacking leucine and containing galactose (top row). Bottom row shows results of counterselection; colonies from the upper row were steaked to plates lacking leucine, containing galactose and 5FOA. **c** Time course of growth in liquid media of the isolated transformants indicated. Error bars reflect standard error SE (standard error = SD//√n); n = 4 for each time point representative of four biological replicates. **d** Growth toxicity assay for scC11 overexpression. The "wild type" laboratory strain BY4741 with an intact chromosomal _RPC11_ gene was transformed with _GAL1_ promoter-driven expression plasmid pYX113 with no inset (ev) or with inserts indicated to the left of the plates. Both plates contained synthetic complete (SC) media lacking uracil, the left plate contained glucose and the right plate contained 2% each of galactose and raffinose. **e** Western blot of whole-cell extracts from strains indicated above the lanes, developed with an antibody raised against _S. cerevisiae_ full-length scC11. Lane 1: _S. cerevisiae_ BY4741; lane 2: yGAKL (_rpc11_Δ, complemented by pRS316-GAL1-spC11; lane 3: yGAKL\SML1a (plasmid shuffle transformant with scC11); lane 4: yGAKL\SML2a: (plasmid shuffle transformant with scNTD-L); lane M: MW markers indicated in kD from an adjacent lane of the same gel. The open triangle to the left of lane 1 indicates a band corresponding to scC11. **f** Northern blot of total RNA, 20 µg (lanes 1, 2) and 10 µg (lanes 3, 4) from cells described in **e**, probed for scC11 mRNA (top panel), spC11 mRNA (2nd panel), and SRP ncRNA (3rd panel). The stable SRP ncRNA provides a size marker as 522 nt RNA, which was also transferred to the top panel. The open triangle to the left of the top panel points to endogenous scC11 mRNA whose estimated size with modest poly(A) would be ~590 nt. The asterisk at the second panel represents a size marker corresponding to ~510 nt. The bottom panel shows the high molecular weight region only of the ethidium bromide-stained gel prior to transfer, with 25S and 18S rRNAs indicated, to serve as quality control for the integrity of the RNA.

(FOA)[63]; viable cells will grow on 5FOA media only after loss of the _URA3_ spC11 plasmid and acquisition of a _LEU2_-selectable plasmid expressing a scC11 construct that can confer its essential function.

After transformation with the _LEU2_-scC11 construct plasmids, multiple individual transformant colonies were streaked on plates permissive for growth with both plasmids and then onto plates containing 5FOA. The top and bottom rows of Fig. 8b show results representative of one of the plasmid shuffle experiments. Multiple transformants of C11, NTD-L, and the linker mutants C11-(Lkmt), NTD-L-(Lkmt) reproducibly yielded viable _S. cerevisiae_ colonies (Fig. 8b, bottom). By contrast, CTD-L and C11-(DE-AA) did not support viability (hints of positive transformants proved false as they could not be propagated).

The NTD-L, NTD-L-(Lkmt), and C11-(Lkmt) transformants reproducibly exhibited slow growth on subsequent plates. Growth in liquid media demonstrated that the central linker DD-AA mutant C11-(Lkmt) was compromised relative to C11, and NTD-L was slower growing with or without these linker mutations (Fig. 8c). To examine the scC11 constructs for toxicity in the presence of chromosomal _RPC11_, BY4741 _S. cerevisiae_ was

transformed with expression plasmids. Serial dilutions onto media that support low-level expression or induce high expression revealed a slow growth defect of C11-(DE-AA) after spotting onto high expression media, manifested as small colony size relative to C11 and the other mutants (Fig. 8d). By contrast, the low expression plates revealed no significant differences in colony size among the constructs.

**The C11 NTD-L mutant exhibits expected expression products**. Antibody raised against full-length scC11 recognized a band in the pYX242-scC11 shuffle transformed yGAKL\SML1a (Fig. 8e, lane 3) that comigrated with a less abundant band in BY4741 (lane 1). A faster band was in NTD-L transformed yGAKL \SML2a (lane 4) but not _rpc11_Δ yGAKL (lane 2), as expected for unreactivity to spC11[46].

RNAs from wild-type BY4741, yGAKL, and its transformants were analyzed by northern blot with probes specific for scC11 and spC11 mRNAs (Fig. 8f). The scC11 probe detected abundant bands of different sizes (Fig. 8f, top panel, lanes 3 and 4), a low-abundance band in BY4741 (lane 1), and no detectable scC11 in yGAKL (lane 2), all as expected. The spC11 probe showed a

relatively abundant mRNA in yGAKL (Fig. 8f, second panel, lane 2) as expected but importantly, no residual spC11 mRNA in yGAKL\SML1a nor yGAKL\SML2a after counterselection (lanes 3 and 4). The third and fourth panels are controls for loading, size, and abundance calibrations. The results in Fig. 8e, f provide evidence in support of the conclusion that the NTD-L confers the essential function of C11 in *S. cerevisiae*.

Additional evidence that NTD alone does not support viability was provided by in vivo TMS (Supplementary Fig. 5). *S. pombe* strain yYH1 produces TMS when C11 mutants are deficient for RNA 3′-cleavage activity[46]. The NTD-L expression construct produced TMS although less than the positive control C11-D90G,R107C[46] (Supplementary Fig. 5, bottom left vs. top right). In these overexpression assays, TMS occurs when a mutant C11 is incorporated into Pol III at the expense of the endogenous subunit[46]. Thus, if NTD could stably interact with Pol III the expectation would be cleavage-deficiency and suppression, but it was inactive (Supplementary Fig. 5, lower left). Also, the CTD alone with the D90G,R107C mutations and without mutations were inactive (far right). Likewise, a F32S-mutated version of the NTD fragment was inactive (Supplementary Fig. 5). That the NTD and CTD lack activity in this in vivo assay emphasizes the specificity of the positive activity conferred by NTD-L. It should also be emphasized that for suppression to occur in this strain, Pol III bearing NTD-L would have to initiate transcription on the suppressor-tRNA gene in order to exhibit a termination-associated RNA 3′-cleavage-deficiency phenotype. Neither the NTD nor the CTD-D90G,R107C produced such phenotype whereas NTD-L did.

## Discussion
The data reported here have advanced our understanding of Pol III mechanisms which include a revised role for C11 CTD-mediated RNA 3′-cleavage, and the conclusion that this cleavage activity is nonessential for viability. The advances came from examination of the C11 subunit by dissection of three activities and their parsing to phases of the transcription process as well as their mapping to its CTD, NTD, or NTD-L. These activities are (i) stimulation of C37/53-dependent termination which was attributed to the NTD, (ii) C37/53-dependent Pol III reinitiation–recycling, attributed to the NTD-linker, and (iii) activation of Pol III intrinsic RNA 3′-cleavage activity which was known to map to the CTD. Similar to TFIIS, the C11-mediated RNA 3′-cleavage activity could reactivate arrested Pol III for elongation[40].

Of the three nuclear RNA polymerase-associated 3′-cleavage factors examined in *S. cerevisiae*, only the one for Pol III is essential[40] (Fig. 8b), whereas Pol II- and Pol I-associated TFIIS (*DST1/PPR2*) and RPA12.2 (*RPA12/RRN4*) are not[52–54]. Thus while Fig. 8b confirms that *RPC11* is essential[40], it shows that its RNA 3′-cleavage activity is not. This now groups Pol III with Pol II, Pol I, and bacterial RNA polymerase, whose intrinsic RNA 3′ cleavage-stimulatory activities are dispensable[64]. However uncovering this commonality further distinguishes the significance of the NTD-L which confers Pol III reinitiation–recycling activity in vitro, as well as the essential function of C11 in vivo. The latter is notable with regard to the nonessentiality of what is considered the homologous Pol I subunit Rpa12.2[52,54], and the relative paucity of sequence conservation in its linker and NTD[65]. Therefore, these conclusions resolve important longstanding biological issues regarding the essential function of C11. The earlier report[40] led to the idea that the RNA 3′-cleavage activity of C11 was an essential function. Our data provide compelling evidence that the essential function is the NTD-L-mediated termination-associated reinitiation–recycling, which is unique to

Pol III among the RNA polymerases. Results of the recycling assays reconstituted with the multiple C11 domains are consistent with this and a model in which this unique activity is essential. As discussed below, the CTD-mediated RNA 3′-cleavage activity is important, it can function in termination and it promotes growth rate, but is nonessential for viability, and likewise is not required for Pol III recycling activity. Thus the collective data provide satisfying and unifying consistency.

**Insight into C11 involvement in Pol III transcription.** It would appear that previous genetic approaches have led to two types of confounding results obtained with C11-(DE-AA) or other cleavage-deficient mutants. Describing how the present report helps resolve may also help advance understanding of Pol III transcription. Prior studies reported that C11-(DE-AA) was unable to rescue viability due to toxicity of overexpression[40]. Overexpression toxicity lethality was also observed in *S. pombe*[46]. Moreover, the C11-(DE-AA) mutations were deemed lethal because the plasmid shuffle experiments used promoter-repressive expression[40]. However, whether the mutant C11 achieved accumulation to the same level as endogenous under those conditions was not reported. Given the collective results including the data reported here the complex issue of whether there may be a titratable level of expression at which the DE-AA mutant could provide the essential C11 function without lethal toxicity remains to be determined.

Notable insight was from the observation that C11-(DE-AA) was not only inactive for cleavage (Fig. 2d) but interfered with termination during C37/53-dependent Pol III reinitiation–recycling (Figs. 5, 6). Moreover, this dose-dependent negative activity of C11-(DE-AA) (Fig. 6 and Supplementary Fig. 4) is in line with the dose-dependent toxicity of C11-(DE-AA)[40]. Although we did not systematically compare constructs for both activities, of all analyzed no other cleavage-deficient mutant exhibited termination interference in the recycling assay nor toxicity (e.g., NTD-L, C11-Lkmt, C11-Δhp) (Figs. 6 and 8d and Supplementary Fig. 4). Thus although not a methodical negative correlation, it would appear specific.

The second type of confounding results prompted the idea that C11-mediated RNA 3′-cleavage may be required for normal termination[46]. This came from a genetic screen of a randomly mutated C11 library that employed a suppressor-tRNA gene architecture with a relatively long, 7T tract terminator in *S. pombe*. The mutants obtained had mutations clustered in each of the four cysteines as well as the hairpin of the CTD that were deficient for RNA 3′-cleavage[46]. (Screening of same C11 library with a different suppressor-tRNA gene architecture that selected for terminator R–T produced distinctive effects that whose C11 mutations clustered in the NTD[37].) Two of the characterized CTD cleavage-deficient mutants produced prolonged 3′-oligo(U) tails on precursor-tRNAs which increased their interaction with the pre-tRNA chaperone La protein leading to high-efficiency production of suppressor-tRNA[46]. Those results added to other data that Pol III intrinsic RNA 3′-cleavage occurs at U residues during termination[66]. Moreover, they provided evidence that C11-mediated RNA 3′-cleavage is a means to modulate the functional outcome of Pol III termination[46]. However, nonessentiality of the CTD, slow growth of the NTD-L mutant, and termination interference by C11-(DE-AA), suggest alternative interpretations and a more integrated model of Pol III transcription.

**An integrated multifunctional-C11 model of Pol III transcription.** C37/53 was shown to stimulate termination by Pol III in the absence of C11[22], which was confirmed using the tailed

template and single-round transcription assays on assembled DNA scaffolds[49]. The latter and additional studies showed that the Pol III-holoenzyme terminates proximally in a 9T tract, whereas Pol III-core lacking C37/53 and C11 is deficient for proximal termination, it extends into the distal 9T tract where it terminates efficiently[49], triggered by the instability of the prolonged (rU:dA) hybrid[39]. The T3-T5 nucleotides in the non-template DNA strand were identified as part of a sequence-specific pause and release element that is recognized by Pol III-holo for termination but not by Pol III-core[33]. DNA scaffolds with a 5T tract limited termination to the proximal region, revealing that C11 increased C37/53-mediated termination reconstituting the holoenzyme mechanism involving a short (rU:dA) hybrid[39]. Moreover, this revealed that this C11 activity was independent of its role in RNA 3′-cleavage[39], reminiscent of its role in the Pol III recycling process[22].

That C11-WT, C11-(DE-AA), and the isolated NTD are comparable in the DNA scaffold termination assay reflects its NTD effect that is C37/53-dependent (Fig. 3). This is consistent with mutants obtained from a termination loss of function screen of mutagenized C11 whose mutations were in NTD amino acids that interact with a conserved extension of C37 that emanates from its C53 dimerization domain[37,41]. The collective data are consistent with a model in which the NTD of C11 contributes a function independent of the linker-and-CTD in Pol III termination. In this model, the C11 NTD stimulates the activity of the C37/53 heterodimer to increase the efficiency by which Pol III recognizes and terminates in a 5T (proximal) terminator, in the context of an (rU:dA) hybrid. The data in Fig. 3 together with cryo-EM structures in elongation mode[41] indicate that such a termination effect by the C11 NTD is dependent on interactions with C37/53 at the Pol III periphery but would not involve interactions transmitted to or from the active center by the C11 linker or CTD.

The integrated model argues that although the NTD of C11 can stimulate C37/53-dependent termination, it is inactive on its own for C37/53-dependent Pol III reinitiation–recycling, which requires the NTD linker or a longer C11 construct (Fig. 5). Thus the C11 linker appears to integrate the otherwise isolated ability of the NTD to stimulate C37/53-dependent termination with Pol III reinitiation–recycling activity. It should be noted that we believe that our data that support this part of the integrated model fits with, supports, and is supported by recent observations of the CTD and linker of human RPC10 in different positions in Pol III ECs and the 3D variability analysis of the corresponding cryo-EM structures[43]. More specifically, structures in which the RPC10 linker-CTD was inside the funnel as compared to outside were accompanied by changes in the heterotrimer initiation subunits and stalk domain and associated with an "open clamp" conformation[43]. Similar positional differences were found for different hPol III structures by others[44]. An integrated model would suggest that as Pol III enters the termination zone the loose RNA–DNA active center-binding site of the EC[41] would adjust in response to termination signal recognition[33] and to the ensuing rU:dA hybrid instability[39]. Movement of the C11 linker-CTD, inside the funnel with the hairpin penetrating the pore is envisioned to further destabilize the active center concomitant with or directly promoting RNA release[43]. Further transition from the heterotrimer clamp close to the open conformation could advance termination by releasing DNA and prepare Pol III for reinitiation by allowing reloading of dsDNA and subsequent DNA strand melting to an open complex[43].

In the revised Pol III model of termination, RNA can be released in the proximal T-tract, dependent on C37/53 but independent of C11-mediated cleavage activity, whereas extension into a distal T tract comes with the risk of arrest[49] and may require an additional rescue activity. This is because the Pol III-holoenzyme, as well as Pol III-core, undergo arrest in (rU:dA) hybrids in long T-tract terminators[33]. We propose that C11-mediated cleavage activity serves to rescue Pol III that fails to terminate in a proximal T-tract and might otherwise falter[66] or arrest in a lengthening (rU:dA) hybrid[39]. Cleavage activity would realign the RNA 3′ end and more importantly reverse arrest to allow spontaneous release associated with (rU:dA)(n) instability[39]. For a terminator proximal region where the degree of (rU:dA) hybrid instability is less than when longer, a previous study suggests that C37/53 alone without the need for C11 can manage RNA 3′-end termination[39]. Accordingly, we suspect that the C11 CTD mutants obtained from the genetic screen that used the long terminator[46] may reflect the rescue activity pathway rather than a requirement for RNA 3′-cleavage per se in normal termination.

## Potential function for the C11-distal linker inside the funnel?

Some of previous biochemistry results are worthy of consideration in light of reported genetic and structural studies. Randomized mutagenesis of *S. pombe* Rpc1 followed by screening for disruption of normal termination identified clusters of single-point mutants in four key regions, (i) the trigger loop and its connected helices and (ii) the bridge helix which participate in enzymatic and translocation mechanisms, and more relevant to the present study, (iii) the extended funnel helices, and (iv) another pair of shorter helices connected by a short loop referred to as the cleft-pore entrance loop (CPEL)[34]. The base of these two pairs of helices forms the entrance to the pore through which the C11-distal linker and CTD pass to the active center. Clustering of termination mutations in the Pol III funnel helices and associated tunnel elements, as well as the CPEL, now provide additional evidence for C11-distal linker and CTD involvement in termination[34] (Supplementary Figs. 6 and 7). Two classes of mutants were obtained, those consistent with debilitating effects on enzymatic function/elongation and those consistent with accelerated elongation[34]. Human Pol III EC structures with C11 CTD inside the funnel allowed us to map those mutations (Supplementary Figs. 6 and 7). Moreover, although yeast Pol III ECs did not contain a visible C11 CTD, they could be used to model the highly conserved linker-CTD of TFIIS[34]. This revealed that the high conservation of amino acid sequence between the human and *S. cerevisiae* Pol III funnel helices (Supplementary Fig. 6) translates to structural conservation, including of the positions of the residues found mutated in the genetic screen (Supplementary Fig. 7e, f).

In addition to the ability to map these mutations, the human Pol III structures show that the distal linker of RPC10 corresponding to the distal linker of scNTD-L construct used in the work reported here may occupy the funnel at the pore entrance (Supplementary Fig. 7e, f). Moreover, the TFIIS distal linker could be modeled in the same near relative position in the *S. cerevisiae* Pol III EC (Supplementary Fig. 7f). It therefore, seems likely that the *S. cerevisiae* C11-distal linker would also maintain a conserved arrangement. Thus, we expect that the C11 NTD-L could access the funnel and have enough range of motion in absence of CTD to sweep into the pore. On the basis of these analyses, the ability to participate in Pol III termination or -associated reinitiation–recycling would not seem difficult to reconcile for constructs lacking a CTD, if all that is necessary is disruption of the funnel pore entrance, such that the distal NTD linker might work as a wedge that pries or nudges the funnel rim or nearby pore entrance to the active center. By contrast, the CTD with its RNA 3′ cleavage activity would be required to rescue arrest per se.

Alternatively, it is possible that the critical structure-function aspects of NTD-L for termination-associated reinitiation–recycling are limited to its 'outside the funnel' activity. In this conformation, the linker of RPC10 had not passed through the restrictive groove entrance at the top of the funnel helices and is instead sharply turned back so the CTD hairpin points at the NTD.

The integrated model proposes that NTD-L is sufficient to support Pol III termination-associated reinitiation–recycling. The model acknowledges that the CTD-mediated RNA 3′-cleavage can provide termination rescue activity and facilitate recycling by resolving arrested Pol III. We expect that while CTD-mediated RNA 3′-cleavage is the preferred rescue mechanism, other mechanisms may act in its absence, including at Pol III arrest at non-terminator sites. Surely, cells survive in the absence of TFIIS and Rpa12.2, the RNA 3′-cleavage stimulatory factors for Pols II and I, and the presumed benefits of resolving arrest[52–54]. Nonetheless, that the CTD can rescue arrested Pol III may likely explain the growth phenotypes of the plasmid shuffle mutants (Fig. 8c). Specifically, the linker mutations in C11-(Lkmt) would limit access of the CTD activities to the active center, presumably including arrest rescue activity whereas NTD-L and NTD-L(Lkmt) are more severely compromised but equally so because they completely lack the CTD.

The invariant residues in the central linker were critical for CTD-mediated RNA 3′-cleavage activity in vitro but not for Pol III reinitiation–recycling (Fig. 7b, c). These linker mutations impair growth rate in full-length C11-(Lkmt) (Fig. 8c) consistent with the idea that C11-mediated cleavage is nonessential but supports growth. Similar growth phenotypes of NTD-L and NTD-L(Lkmt) suggest that linker access to the funnel is irrelevant to the growth of these NTD-L mutants. This further suggests that the essential function of the NTD-L may not be critically impaired by the DD-AA mutations if not connected to a bulky CTD.

## Methods

Yeast strains are listed in Table 1. A complete list of the oligo-DNAs and -RNAs used is provided in Supplementary Table 1. These were purchased from Integrated DNA Technologies or Eurofins Genomics. Nucleotide triphosphates (NTPs) were from Thermo-Scientific. Ni-NTA beads were from Qiagen.

**Purification of S. cerevisiae C11 protein and its derivatives**. The C11 and C11(D91A-E92A), as well as additional C11-derivatives cloned in the BamHI/XhoI sites of pGEX-4T-1 plasmid, were purified as described[39]. The additional derivatives are C11-(D52AD53A), C11-Δhp (Δ81-95), NTD (1–36), NTD-L (1–70), NTD-L-(D52AD53A)-(1–70), CTD-(60–110), and CTD-L-(38–110). These same BamHI-XhoI fragments were subcloned into pYX242 for in vivo assays. These fragments of C11 were generated by PCR. The pGEX-4T-1 plasmids containing the C11 inserts were transformed, expressed in Rosetta (DE3) pLysS cells with 100 uM Isopropyl -D-1 thiogalacto pyranoside (IPTG) for 16 h at 18 °C. Cells were lysed in lysis buffer (50 mM K-HEPES, pH 7.8, 500 mM NaCl, 5% glycerol, 10 mM 2-mercaptoethanol, 1%Triton X-100, and protease inhibitors) and then centrifuged at 15,000×g at 4 °C for 30 min. The supernatant was incubated with glutathione–Sepharose (GE Healthcare), equilibrated and washed with lysis buffer,

and then with wash buffer (WB; 20 mM K-HEPES, pH 7.8, 200 mM NaCl, 10% glycerol, 10 mM 2-mercaptoethanol). Protein was eluted by digestion with thrombin (Biopharm laboratories) at 4 °C overnight in WB. The thrombin was inactivated by adjusting the solution to 2 mM benzamidine–HCl, and the purified proteins were stored at −70 °C.

**Purification of C37/53 heterodimers**. C37/53, C37*/53 (alanine substitutions at positions 226–230), and C37**/53 (deletion of amino acids 226–230) recombinant heterodimers were purified as described[39]. Both subunits were co-expressed in bacteria. pET28-nH6TEVC53 was cotransformed with pET21-nFLAGC37 or mutated versions thereof into Rosetta (DE3) pLysS and the cells were plated on LB media containing ampicillin (100 g/ml) and kanamycin (50 g/ml) and induced with 0.5 mM IPTG. Cells were lysed in lysis buffer (50 mM Na-HEPES, pH 8, 200 mM NaCl, 5% glycerol, 10 mM 2-mercaptoethanol, and protease inhibitors), collected by centrifugation at 15,000×g, the supernatant was collected and passed through Ni-NTA resin (Qiagen). The complexes were eluted with 20 mM K-HEPES, pH 8, 200 mM NaCl, 10 mM 2-mercapto- ethanol, 10% glycerol 300 mM imidazole. Peak fractions were pooled, a total of 40 units of Turbo TEV protease (Eaton Biosciences Inc., San Diego) was added, and dialyzed against elution buffer lacking imidazole. Samples were again passed through Ni-NTA to retain unwanted tagged fragments/undigested proteins and the flowthrough was passed through an SP Sepharose (GE Healthcare) column equilibrated with dialysis buffer using the ÄKTA purifier system (GE Amersham). Peak fractions were pooled, and buffer (20 mM Tris-Cl, pH 8, 200 mM NaCl, 10 mM 2-mercaptoethanol, 10% glycerol, and protease inhibitors) was exchanged using a PD10 column (GE Healthcare) using the ÄKTA system. The C53/37 complexes were applied to a Q-Sepharose (GE Healthcare) column and eluted with 300 mM NaCl.

**Purification of Pol III**. Pol III-holo was purified as described from S. cerevisiae yZN16, carrying an N-His-FLAG-tagged RPC128 (C128) gene; Pol III-core was purified from yGAKL carrying N-His6-FLAG3-tagged C128 (RET1) in which Rpc11 was deleted and complemented with S. pombe C11[28,49]. Briefly, 60 g of cells pellet was lysed by bead beater in ice-cold lysis buffer (40 mM Na-HEPES, pH 7.8, 5% glycerol, 10 mM 2-mercaptoethanol, 0.5 M NaCl, 7 mM MgCl2, and protease inhibitors). The supernatant was collected after centrifugation at 15,000×g for 30 min and then ultracentrifugation (Beckman) at 100,000×g in a 60 Ti rotor for 1 h at 4 °C. The top layer was recovered (S100), avoiding the murky stuff above the pellet. The S100 was incubated with Ni-NTA resin (Qiagen) at 4 °C for 2 h. Pol III was eluted with 20 mM Na-HEPES, pH 8, 20% glycerol, 500 mM NaCl, 10 mM 2-mercaptoethanol, 7 mM MgCl2, 100 mM imidazole, and complete protease inhibitor (Roche). Small aliquots were stored at −70 °C. For the assays, units of activity for Pol III-holo and Pol III-core is that which give near equal amounts of total RNA products from ECs assembled on DNA scaffolds with a 9T terminator, as seen for example in Fig. 3a/b and 8 in ref. [39].

**EC assembly for transcription termination assays and cleavage assays**. EC assembly on scaffolds and transcription reactions were performed as described[39]. Pol III with a His-tag on Rpc2 was immobilized on magnetic Ni-NTA beads (Thermo-Scientific) and washed with EC buffer (EC buffer; 20 mM Na-HEPES, pH 8, 3 mM β mercaptoethanol, 5% glycerol, 0.1 mg/ml bovine serum albumin, 100 mM NaCl). Template (T) DNA (1 μM of SMO77) and 5′ 32P-labeled RNA were annealed in ECB by heating to 55 °C and rapid cooling to room temperature and incubated with immobilized Pol III and after 10 min incubation at room temperature, 2 μM nontemplate (NT, SMO27X). DNA was added and incubated for 10 min followed by 3washes. Now the EC assembly is ready for transcription. For all experiments, equal amounts of Pol III-holo and Pol III-core transcription activities were used. A typical 1× reaction contained 25 μl of beads. For some experiments, a larger batch of ECs was prepared, washed then aliquoted into equal smaller volumes for subsequent analysis. For reconstitutions of Pol III, recombinant proteins were incubated with the ECs for 15 min prior to NTP addition. A total of 500 μM NTPs containing 7 mM MgCl2 was added and incubated for

### Table 1 Yeast strains used in this study.

| Strain name | Genotype | Description | Reference |
|---|---|---|---|
| yNZ16 (S. cerevisiae) | N-His-FLAG-RPC128 | Pol III-holo purification | 49 |
| yGAKL (S. cerevisiae) | N-His-FLAG-C128, rpo31D::KanMX, pRS313gu-nHA-C160, rpc11D::NatMX, pRS316gu-spC11mvq. | Pol III-core purification. Also used as a recipient for plasmid shuffle | 49 |
| ySHA24 (S. pombe) | h- ade6-704 ura4-D18 leu1-32::tRNApSer4T-leu1+ | Red–white tRNA-mediated suppression | This study |
| yRS6a (S. pombe) | ySHA24 strain containing nLuc-ffly-Luc dual-luc with UGA stop codon within the nLuc ORF. | Used for tRNA-mediated opal-suppressible nanoluciferase assay | This study |
| BY4741 (F729) | MATa his3Δ1 leu2Δ0 met15Δ0 ura3Δ0 | Standard wild-type S. cerevisiae strain | 69 |
| SML1a (S. cerevisiae) | yGAKL strain containing pYX242 WT scC11 | Strain created by plasmid shuffling method | This study |
| SML2a (S. cerevisiae) | yGAKL strain containing pYX242 scC11-NTD-L | Strain created by plasmid shuffling method | This study |
| yYH1 (S. pombe) | h- ade6-704 ura4-D18 leu1-32::tRNApSer7T-leu1+ | Red–white tRNA-mediated suppression | 46 |

10 min at room temperature followed by separation of released from bound by the magnet and immediate phenol extraction/ethanol precipitation. Samples were dissolved in formamide dye, resolved on 15% sequencing gels, dried, exposed to PI screen, and visualized using Typhoon (GE Healthcare).

A $^{32}$P 5′-labeled 10-nt RNA annealed to the DNA template (T) strand with 5 As in the termination zone (TZ) assembled with Pol III-core and a nontemplate (NT) DNA strand (5Ts in the TZ) is prepared and washed. The ECs are aliquoted to 1.5-ml tubes, components are added, reactions are initiated and incubated for 10 min and then immediately stopped upon separation of the supernatant, which represents the terminated or released (R) RNA fraction, and the bound (B) RNA fraction that remains associated with Pol III. The purified RNA products are separated on urea polyacrylamide gels which are processed, the $^{32}$P-RNA quantified, and termination efficiency at the 5T terminator is calculated[39].

For cleavage assays, the EC was assembled as described above with a 2-nt 3′ overhanged RNA primer employed and used template AGO16X, RNA primer RNA6, and NT AGO 7X. And incubated with the presence or absence of WT C11/ derivatives of C11 for 15 min at room temperature followed by the addition of 7 mM MgCl$_2$ and incubated for 15 min; the reaction was stopped and samples loaded in the gel.

Transcription termination efficiency (%) was calculated as follows. Transcription termination efficiency = (RNA released at TZ/total RNA) × 100. Total = RNA released in TZ + released RT RNA + bound at TZ + bound at RT). TZ refers to terminator zone (TZ) and RT refers to read-through. Phosphoimages were analyzed by using Multigauze V3.2 software (Fuji)[39].

**Preparation of S. cerevisiae cell extract[67].** Cells were grown in YPD media (1% w/v yeast extract, 2% w/v peptone, 2% w/v dextrose) to a density at 650 nm of 3.17 g of cells were harvested by centrifugation, washed with chilled distilled water, and resuspended into 25 ml lysis buffer (200 mM Tris-C1 (pH 8.0), 10% v/v glycerol, 10 mM MgCl$_2$, 2 mM DTT, 1 mM phenylmethylsulfonyl fluoride (PMSF) and EDTA free protease inhibitor (Roche) and poured into 60-ml small chamber of bead beater cell disrupter. An equal volume of 0.445-mm diameter acid-washed glass beads (Sigma) sufficient to exclude all air from the chamber was added to the chamber. The unit was assembled and placed in an ice jacket containing an ethanol-ice slurry. Cells were lysed by homogenization for 9–10 min total time, in 30-s bursts with 4 min cooling time between bursts. Tlysate was decanted from the glass beads, and 1 mM PMSF and 2 mM DTT were re-added, respectively. 0.4 M (final concentration) ammonium sulfate was added in the lysate from 4 M stock ammonium sulfate (pH 7.9). The extract was incubated on ice for 30 min before centrifugation at 15,000×g for 60 min. The supernatant was collected and centrifuged at 100,000×g (rotor 60 Ti) again for 1 h. The supernatant was collected. The proteins in the supernatant were precipitated by adding very slowly ammonium sulfate (0.35 g per ml of lysate) and kept on ice for 30 min then centrifuged to precipitate the proteins. The precipitated proteins were dissolved in a minimal volume (10 ml) of Buffer C (20 mM HEPES KOH, pH 7.9, 20% glycerol, 0.2 mM EDTA, 2 mM DTT, 1 mM PMSF) and desalted by using a S200 column (2.5 cm diameter and 12 cm height) as described[68] and elution was collected after 1.5–2× of the load volume passed from the column by gravity flow.

**Promotor-dependent Pol III recycling assays.** Preparation of Pol III-depleted initiation complexes was as follows, prepared in batch, calculated per reaction. (1) 200 ng of biotinylated PCR-derived sup4e DNA, immobilized on 30 ul magnetic streptavidin beads, incubated with 30 μl transcription buffer (TB; 40 mM Na-HEPES, pH 8, 3 mM DTT, 5% glycerol, 0.1 mg/ml bovine serum albumin, 7 mM MgCl$_2$, 100 mM NaCl) containing 10 μl of Txn extract for 20 min at RT. (2) The supernatant was replaced with 25 μl TB containing 0.5 mM each nonradioactive NTP with heparin at 0.03 mg/ml, incubated for an additional 3 min at RT. (3) The complexes were washed once with 50 μl of TB containing 500 mM NaCl and twice with 50 μl of TB. (4) Complexes were then used for subsequent transcription by purified Pol III-holo or Pol III-core.

The subsequent transcription assay was as follows. Pol III-holo or Pol III-core was added to the freshly prepared extract-derived Pol III-stripped complexes. After 20 min of incubation, the supernatant was replaced with 20 μl of TB, and 5 μl of $^{32}$P-NTP mix was added (TB containing 0.5 mM ATP, GTP, CTP; and 0.02 mM UTP with 1 μCi $^{32}$P-αUTP (1 Ci=37 GBq). Reactions were stopped after 30 min of incubation at RT by the addition of 50 μl acidic phenol. Total RNA was purified by ethanol precipitation, resuspended into 15 μl RNA loading buffer (Invitrogen), and analyzed by 8% Urea-PAGE.

**Cloning of S. pombe C11 and its derivatives.** S. pombe WT C11 and C11-(D90G, R107C) in pRep4X were available[46]. C11 NTD-L(1–60) and CTD-L(36–109) were PCR amplified and cloned in pRep4X under BamHI and XhoI-restriction sites.

**Plasmid shuffle.** S. cerevisiae yGAKL strain (a gift from George Kassavetis, UCSD) from which the essential chromosomal RPC11 gene was deleted, is hetero-complemented by S. pombe C11 expressed from the centromeric plasmid pRS316 (URA3) under control of the GAL1 promoter. This strain is viable when galactose is the sole carbon source. WT scC11 and its derivatives cloned in pYX242, 2 μm (LEU2), were transformed into yGAKL strain and plated on SC (synthetic

complete) media without Leu and Ura +galactose plates. Multiple colonies from each transformant plate were streaked on SC w/o Leu +galactose plates and these colonies were then streaked on SC −Leu +Galactose +5FOA plates. The URA3 pRS316 containing S. pombe C11 will be counter-selected against by 5FOA (see the text).

**Growth toxicity assays.** S. cerevisiae strain F729 (a gift from Tom Dever, formerly BY4741[69]) was transformed with pYX113 plasmid which has a GAL1 promoter[70] driving expression of the different scC11 constructs, and plated on SC media lacking uracil. Colonies were inoculated in liquid media (SC without uracil) for overnight growth at 30 °C. From the overnight culture, cells were diluted into fresh SC media without uracil and incubated at 30 °C. Logarithmically growing cells at an OD$_{600}$ of 0.6 were spotted at a cell concentration of 1.0 OD/ml and tenfold dilutions thereof; 5 μl of each was spotted onto SC + glucose w/o uracil plates and SC + galactose +raffinose w/o uracil plates and grown at 30 °C for 3 days.

**tRNA-mediated suppression.** Expression plasmid pRep4X-mediated suppression using the red–white assay based on suppressible ade6-704 in S. pombe strains harboring integrated suppressor-tRNA genes was as previously described[57] (reviewed in ref. [61]). Strain ySHA24 was transformed with pRep4X containing different derivatives of C11, Rpc1 or the empty vector and other and plated onto EMM-complete media lacking uracil with limiting adenine (10 mg/L). Representative transformant colonies were streaked and subsequently transferred to liquid media, grown overnight, diluted, grown logarithmically to OD$_{600}$ of 0.6. Cells were spotted onto plates as above and grown at 32 °C for color development.

For the dual-luciferase assay, a construct with the following gene arrangement; NanoLuciferase-KanR-firefly Luciferase, was inserted by homologous recombination into S. pombe Chr I between SPAC6F6.11c and SPAC6F6.12[71]. Each of the genes in the construct is an independent transcription unit with its own promoter, Kozak sequence, and terminator. The nano-Luc gene was mutated to replace a TCC serine codon with a TGA premature stop codon that is suppressible by the opal suppressor-tRNA and its derivatives[61]. The pRep4X-mediated suppression was as above. Cells were collected and used to make an extract from which dual-luciferase activities were measured.

**Liquid growth assays.** S. cerevisiae strains having different scC11 constructs in pYX242 were grown for 24 h at 30 °C, 250 rpm shaker in 5 ml synthetic complete (SC) media without leucine +galactose liquid media. After 24 h, the cultures were diluted to 0.05 (OD$_{600}$) in 25 ml of the same media in a 250-ml flask for each. At each time point, OD$_{600}$ was measured as indicated.

**Northern blot assays.** Total RNA was isolated using hot phenol. In short, 50-ml cultures grown from an OD$_{600}$ of 0.1–0.4 were harvested, washed with water, and resuspended in 300 μl TES buffer (10 mM Tris-Cl pH 7,10 mM EDTA, 1% SDS). In total, 300 μl acidic phenol (Invitrogen) was added and incubated at 65 °C for 60 min, with vortex every 10 min. To the samples, 300 μl chloroform was added and centrifuged. The supernatant was extracted twice with acid-phenol/chloroform and once with chloroform before ethanol precipitation. The precipitated pellet was resuspended into 50 μl water, quantified, and mixed with loading buffer. In all, 10 and 20 μg of RNA was resolved on 1.8% agarose-formaldehyde gel and transferred to a GeneScreen-Plus membrane (PerkinElmer). After UV cross-linking and vacuum-baking at 80 °C for 2 h, the membrane was prehybridized in 6× SSC, 2× Denhardt's, 0.5% SDS, and 100 μg/ml yeast RNA for an hour at the hybridization incubation temperature (Ti). 5′-end-labeled $^{32}$P-oligo DNA probes complementary to scC11 mRNA, SCR1 RNA, and spC11 mRNA were incubated at the Ti overnight and washed. The blot was stripped and examined for probe removal prior to rehybridization.

**Western blotting.** In total, 50 ml of 0.5 OD$_{600}$ cells were harvested, washed, and suspended into lysis buffer (50 mM Tris-HCl pH 7.5, 150 mM NaCl, 2 mM EDTA, 0.5% Triton X-100, and protease inhibitor Cocktail). In all, 40 μg protein/lane was separated by SDS-PAGE on 10–20% polyacrylamide gels and transferred onto the nitrocellulose membrane (GE Healthcare Life Sciences). Membranes were blocked in 1% (w/v) skimmed milk in PBS for 60 min at room temperature and incubated with primary antibody at 1:1000 in PBS-T buffer overnight at 4 °C. Membranes were washed several times in PBS-T, incubated with fluorescent dye-conjugated secondary anti-rabbit Ab (Li-Cor, Lincoln, NE, catalog #P/N 926-32211) in blocking buffer (PBS-T) at 1:10,000 and visualized using Li-Cor system. The custom-made anti-scC11 Ab was raised in rabbit against full-length S. cerevisiae C11 and affinity-purified against the same full-length protein by GenScript USA, Piscataway.

PyMOL molecular graphics software v2.0.1 was used for structure representations. The S. cerevisiae Pol III EC (PDB 5FJ8) with the CTD-Linker of yeast TFIIS modeled in, as reported previously[34], was a kind gift from N Hoffman and C. Muller (EMBL, Heidelberg).

**Statistics and reproducibility.** Each and every experiment depicting results in any form and including all that are considered representative have been repeated

independently with similar results a minimum of two times. We note that a minimum of two independent repeats with similar results was atypical for many of the most relevant samples/constructs tested as part of a series of samples/constructs. More common was that most of the individual samples/constructs were tested four or more times with similar results. Moreover, multiple experiments performed as biological replicates with similar results were then repeated as biological replicates using a different batch of purified proteins, again producing similar results.

## Data availability

The data that support this study are available from the corresponding author upon reasonable request. Source data are provided with this paper.

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

## Acknowledgements

We thank M. Palangat, D. Price, and members of the Maraia Lab for discussion, and M.P. and A. Kessler for helpful comments. We thank R. Moir for advice on yeast transcription extract preparation, T. Dever for BY4741 (F729) cells, A. Vindu for pYX242. We thank Linh Pham for early work on luciferase construction, and S. Gaidamakov, A. Khalique, and other lab members for technical and other help during the COVID pandemic. This work was supported by the Intramural Research Program (HD000412 PGD) of the *Eunice Kennedy Shriver* National Institute of Child Health and Human Development, National Institutes of Health. The funders had no role in study design, data collection and interpretation, or the decision to submit the work for publication.

## Author contributions

S.M. and R.M. conceived experiments, analyzed all data, wrote and edited the paper. S.M. performed all transcription assays and performed or oversaw all other experiments. S.H. constructed the suppressor-tRNA allele in and characterized the *S. pombe* ySHA24 strain with supervision from R.M. R.S. designed, constructed, and characterized the nanoluciferase-firefly dual-luciferase reporter and the *S. pombe* yRS6A strain with supervision from R.M. S.C. made some of the plasmid shuffle constructs, performed, and isolated initial NTD-L transformants with supervision from S.M. and also made some constructs used for and performed TMS assays with supervision from S.M. and R.M.

## Competing interests

The authors declare no competing interests.
