## [Peer Review File · Nature Communications]

REVIEWER COMMENTS

Reviewer #1 (Remarks to the Author):

In their manuscript, Mishra et al. analyze the role of the essential RNA polymerase III-specific subunit C11 in transcription termination, RNA 3'-cleavage and reinitiation-recycling using biochemical and in vivo assays. Based on their experiments, the authors assign distinct activities to the different parts of C11. Namely, the C11 N-terminal (NTD) and C-terminal domains (CTD) enhance termination and RNA 3' cleavage activity, whereas the NTD-linker (a construct comprising the NTD + the linker connecting NTD and CTD) is required for the Pol III-specific reinitiation-recycling activity. So far, 'reinitiation-recycling' or 'facilitated reinitiation' by Pol III had been poorly understood. The manuscript by Mishra et al. provides important insights of the critical role of subunit C11 in this process. I recommend publication in Nature Communications once the different points listed below have been addressed:

Major points:

- 1.) Protein gels confirming the purity and overall sample quality (and differences in size) of the purified C11 constructs should be shown in Fig. 2 or in the supplement.
- 2.) Throughout the paper, the authors compare different constructs to Holo Pol III as well as to the Core complex complemented by C11 and C37/53. Consistently, the reconstructed Core+C11+C37/53 complex has lower activity compared to the Holo complex. The authors should discuss the reason for this discrepancy.
- 3.) Fig 2C shows an RNA cleavage gel. Unfortunately, the gel is not of the same quality as Fig 5D in the same manuscript. Cleaved RNA products are not well separated from the labeled RNA primer, which makes it difficult to judge to what extent the overhang has been cleaved off. In addition, construct CTD is not shown in Fig 2C, although it is later shown in Fig 2D.
- 4.) Fig 2C and 3C show error bars for only 2 replicates.
- 5.) What does in Fig 2D the Y-axis label "fraction of cleavage products" mean? To which signal is this a relative fraction? In Fig S1 for WT C11 no concentration of 400 nM is shown. Where did the authors get the value for this concentration in Fig 2 D?
- 6.) Fig 8E and F are mostly quality control panels. I suggest to move them to the Supplemental Information and to shorten the description of Fig 8 E/F in the text. The following paragraph describing the erroneous NTD constructs could be also shortened.
- 7.) A key finding of the paper is that the RNA-cleavage activity of C11 is not essential because a C11 variant that lacks the catalytically active C-terminal part is able to rescue growth in plasmid shuffling experiments. However, I still wonder why rescue cannot be achieved using the C11_DE-AA mutant? The authors want to discuss this point in more detail.

8.) For the cleavage assays, no C37/C53 was used, which is puzzling at first because it also attributes to the binding interface with the C11 NTD. If I understood correctly, the authors later on in that paragraph argue that C37/C53 was omitted intentionally to assess direct binding of C11 to the Pol III core. I suggest mentioning that fact earlier. In addition, the manuscript might benefit from repeating that experiment with and without C37/C53 to compare activities of WT and mutant C11 constructs in absence and presence of C37/C53.

9.) Fig S5 showing the structure of hPol III in different C11 CTD positions should be revised as it is extremely difficult to recognize the features of importance (such as the C11 linker).

Minor points:

1.) Fig 3C is missing the label "C11" below "C37/53" and the 1st "-" in the row.

2.) Fig 8A is not referenced.

3.) In the introduction, the structure of hPol III having the C11 CTD inserted into the funnel domain is mentioned twice in two separate paragraphs.

4.) In the introduction, the essential C11 function are only briefly mentioned at the end and should be explained in greater detail (including primary citations).

5.) Figure 5: label (A) is shown in two panels. The second (A) in the figure on the bottom left should be changed to (B). The (B) in the right panel should be deleted.

6.) On p13, lane 6, the Fig. reference is missing the actual figure number.

7.) At the bottom of p14/beginning of p15, the authors refer to northern blotting experiments but don't refer to the respective figure (I assume Fig 8F?).

Reviewer #2 (Remarks to the Author):

General Comments

This manuscript from Mishra and colleagues evaluates domain-specific functions of the small, but important C11 subunit of RNA polymerase 3. Using a blend of biochemical and genetic assays, this manuscript assigns unique functions to the N-terminal domain, the linker, and the C-terminal domain of C11.

RNA polymerase 3 is unique in its ability to reinitiate transcription immediately after termination. There is a robust history/literature focused on the roles of individual subunits in this activity as well as the unique termination mechanism presented by the enzyme.

Here, Mishra and colleagues show that 3' RNA cleavage is in fact not essential for RNA polymerase 3, but that termination and recycling are. This conclusion refines previous interpretation of genetic data.

In general, the story is interesting and precise. The findings build on a long history of work with this enzyme, and the lab is clearly expert in the details. Related to that perceived strength, however, it is hard to conclude that this manuscript will appeal to a broad readership like that of Nature Communications. As this study refines many previous interpretations (based on genetics and biochemistry), it seems more appropriate for a specialized journal.

Specific Comments

1. Many (most) of the data are representative and very little quantification or evaluation of reproducibility is displayed or described. Figure 4 is of particular concern since the slight lag in product is interpreted as an effect on binding. Although this interpretation is reasonable, it is not justified without consideration of variability in the results. In short, error analysis and reproducibility is not clear throughout.
2. The timing of experiments should be more clearly explained in the figure and/or legend. The methods section is pretty clear, but some of the assays are difficult to interpret without clear description of timing.
3. Where is CTD alone in figure 2C?
4. Overall the results section is difficult to follow. It is written such that an expert in RNA polymerase 3 might understand and evaluate impact, but it is very difficult for someone outside of the field.
5. The Discussion section is not as well refined grammatically, and should be edited.

A brief summary of new data or other elements in the revised manuscript in response to Reviewers' comments:

- Fig. 2: Added new panel b, coomassie gel showing C11 protein constructs as suggested.
- Fig. 2: Added lanes 13-14, panel d. Also revised image (decompressed gel) to resolve separation as suggested.
- New panel b of Supplementary Fig. 1, linear regression analysis of data in Fig. 2d with trendlines and R² values.
- Fig. 3: Added a new element to panel a, a schematic timeline description of experimental design.
- Fig. 3: Added *p*-values to panel c.
- Fig. 4: Added a new element to panel a, schematic timeline description of experimental design.
- New Supplementary Fig. 3, regression analysis of data in Fig. 4c with trendlines and R² values.
- Revised Supplementary Fig. 5 as suggested, now Supplementary Fig. 7 preceded by new Supplementary Fig. 6, a multiple sequence alignment.

Reviewer #1 (Remarks to the Author):

In their manuscript, Mishra et al. analyze the role of the essential RNA polymerase III-specific subunit C11 in transcription termination, RNA 3'-cleavage and reinitiation-recycling using biochemical and in vivo assays. Based on their experiments, the authors assign distinct activities to the different parts of C11. Namely, the C11 N-terminal (NTD) and C-terminal domains (CTD) enhance termination and RNA 3' cleavage activity, whereas the NTD-linker (a construct comprising the NTD + the linker connecting NTD and CTD) is required for the Pol III-specific reinitiation-recycling activity. So far, 'reinitiation-recycling' or 'facilitated reinitiation' by Pol III had been poorly understood. The manuscript by Mishra et al. provides important insights of the critical role of subunit C11 in this process. I recommend publication in Nature Communications once the different points listed below have been addressed:

•Author response: We thank the Reviewer for careful reading of our paper, analysis of the data, interpretation in context and supportive conclusions. We believe that this has led to a substantially improved manuscript. Some figure panels have been relabeled in the revised version (d vs. c etc.) and will be referred to as such below.

Major points:

1.) Protein gels confirming the purity and overall sample quality (and differences in size) of the purified C11 constructs should be shown in Fig. 2 or in the supplement.

•Author response: We have added the coomassie-stained gel showing the purified C11 protein constructs as a new panel Fig. 2b, as suggested, and note it in the revised manuscript (p. 7). We note here that although a co-purifying contaminating band of ~25 kDa is present at comparable levels in the NTD-L and CTD-L preparations, its identity was not pursued because it was apparently not deterministic of any activity investigated.

2.) Throughout the paper, the authors compare different constructs to Holo Pol III as well as to the Core complex complemented by C11 and C37/53. Consistently, the reconstructed Core+C11+C37/53 complex has lower activity compared to the Holo complex. The authors should discuss the reason for this discrepancy.

•Author response: This is an interesting issue which we welcome the opportunity to address here and more succinctly in the revised manuscript. The Pol III-holo complex showed more activity than the core complex +C11 and C37/53 in the simple termination assay using the DNA scaffold (Fig. 3) as well as in the promoter-dependent recycling assay (Fig. 4), using comparable amounts of "activity" of each preparation that was previously defined. Units of activity for holo and core is that which give near equal amounts of total RNA products from ECs assembled on DNA scaffolds with a 9T terminator as for example in Fig. 3A/B and Fig. 8 in Mishra and Maraia, 2019. We added this unit definition to the Methods (p. 24, bottom).

That there is relatively more recycling activity associated with Pol III-holo than with Pol III-core +[C11+ C37/53] is interesting. Previously, it was noted using scaffolds that -holo performs with some qualitative differences relative to -core +[C11+C37/53] although these were subtle differences in sensitivity to C11 activity (see Fig. 6 in Arimbasseri and Maraia, 2013 Mol Cell Biol). The Fig. 3C scaffold assay results using a 9T terminator in Arimbasseri and Maraia (2015) shows that a difference between -holo and reconstituted -core components is a lower TE and more arrest when C11 is the only protein absent. Data in

Fig. 7A & B in Mishra and Maraia 2019 NAR provided evidence that C37/53 can temper the cleavage activity of C11 in direct cleavage assays (using overhang substrate, this is also related to point 8 below). Finally, (the heading of) Fig. 8 in Mishra and Maraia 2019 indicates that C37/53 plus C11 do not faithfully reconstitute Pol III-core for termination, evidence of differences between Pol III-holo and reconstituted Pol III-core. The penultimate section of the Discussion of Mishra and Maraia 2019 ended noting that addition of C11+C37/53 to Pol III-core does not fully reconstitute Pol III-holo activity, and leaves open the possibility that post-translational modifications may account for such discrepancies. We suspect that a reason why Pol III-holo exhibits more recycling activity is because it is unincumbered by C11 activity relative to reconstituted Pol III-core. We added a more succinct version of the above as the last part of the Discussion of the revised manuscript. “It is tempting to suspect that Pol III-holo exhibits efficient recycling because its native C11 and C37/53 function more cooperatively than the recombinant proteins in unincumbered transitions between the different phases of the transcription cycle.” We thank the Reviewer for the opportunity to discuss this interesting aspect of the data.

3.) Fig. 2C shows an RNA cleavage gel. Unfortunately, the gel is not of the same quality as Fig. 5D in the same manuscript. Cleaved RNA products are not well separated from the labeled RNA primer, which makes it difficult to judge to what extent the overhang has been cleaved off. In addition, construct CTD is not shown in Fig. 2C, although it is later shown in Fig. 2D.

•**Author response:** We thank the Reviewer for pointing out the two problems with the RNA cleavage gel, now Fig. 2d. The appearance of poor separation of cleaved products (CP) from the labeled RNA primer as correctly noted was in part due to erroneous vertical compression of the gel picture during image processing. This was corrected in revised Fig. 2d by undoing the compression to reveal better separation of the cleaved products (CP).

With regard to the additional point, we added data using construct CTD as lanes 13 & 14, to Fig. 2d and appropriately noted it in the legend.

4.) Fig. 2C and 3C show error bars for only 2 replicates.

•**Author response:** Yes, this is correct but we are confident in the significance of the data because they have been reproducible in additional experiments not shown and because the error bars in these experiments, Fig. 2c and 3c, are quite small. To address concern regarding significance of the data in Fig. 2c we subjected them to further analysis, shown in new Supplementary Fig. 1b which revealed that the R^2 values after linear regression analysis of the trendlines fit to the plotted data were quite good at ≥ 0.95 . We refer to the linear regression analysis in Supplementary Fig. 1b in the Results section of the revised MS (p. 7, bottom).

With regard to Fig. 3c, in addition to reproducibility in multiple experiments noted above and small error bars for each point, we added p -values to the Fig. 3c graph. We noted the p -value derivation by a two-tailed paired Students t-Test in the legend to Fig. 3c (p. 35).

Although this reviewer did not comment on error bars in Fig. 4c we realized that some of these were so small they were not noticeable as such, and were not referred to/described in the figure legend. Therefore, in addition to revising the figure legend accordingly we subjected the data in Fig. 4c to regression analysis. Those new data are in Supplementary Fig. 3 and referred to in the revised MS (p. 9, bottom). It revealed that the R^2 values for the lines fitted to the plots were quite good, the best fit for all of the curves was by applying the 2nd order polynomial regression yielding R^2 values of 0.9972 to 0.8266. The second best fit was logarithmic with R^2 values of 0.988 to 0.7895 and with the same hierarchical pattern among the samples, as noted in the legend to Supplementary Fig. 3.

5.) What does in Fig. 2D the Y-axis label “fraction of cleavage products” mean? To which signal is this a relative fraction? In Fig. S1 for WT C11 no concentration of 400 nM is shown. Where did the authors get the value for this concentration in Fig 2 D?

•**Author response:** We thank the Reviewer for noticing and pointing out the incompletely defined Y-axis. We have fixed this by adding the revised Supplementary Fig. 1a lower image, and also in the revised Fig. 2e legend as “% cleavage product (CP) = $CP/(BEC + CP) \times 100$.” (p. 34)

6.) Fig. 8E and F are mostly quality control panels. I suggest to move them to the Supplemental Information and to shorten the description of Fig. 8 E/F in the text. The following paragraph describing the erroneous NTD constructs could be also shortened.

•**Author response:** We have considered these suggestions. We wish to emphasize that although we agree that Fig. 8e/f are quality controls, they provide important support of the conclusion that the NTD-L of C11 confers the essential function of C11 in *S. cerevisiae* which might otherwise be questioned because of the complex genetic technique used, plasmid shuffle and the surprising results obtained because they challenge the longstanding belief that resulted from previous plasmid shuffle experiments (Chedin et al., 1998) from the same premier yeast Laboratory that invented the plasmid shuffle technique a few years earlier (ref 62, Mann et al., 1987). Prior data that led to confounding results on C11, dose-dependent toxicity of C11-D91A,E92A and lethality of the same point mutated ORF under promoter repressive conditions were reported as data not shown by Chedin et al., and this lethality was vaguely claimed as data not shown for C11-E92H by Alic et al. For these reasons including the history of confounding results obtained by similar approach, we believe it would be prudent and prefer to include these data as primary figure elements rather than supplementary.

The text describing the results in both panels were shortened as suggested. Also as suggested the next paragraph thereafter was shortened. Overall this section is shorter by >25% in the revised MS.

7.) A key finding of the paper is that the RNA-cleavage activity of C11 is not essential because a C11 variant that lacks the catalytically active C-terminal part is able to rescue growth in plasmid shuffling experiments. However, I still wonder why rescue cannot be achieved using the C11_DE-AA mutant? The authors want to discuss this point in more detail.

•**Author response:** We thank the Reviewer for raising the important possibility, that it may be possible to achieve rescue by a C11_DE-AA mutant. To best accommodate this together with this Reviewer's suggestion below that the essential C11 function should be explained in greater detail and earlier, the Introduction and Discussions were somewhat reorganized. We wrote in the revised manuscript (p. 18, top) Discussion, "Given the collective results including the data reported here the complex issue of whether there may be a titratable level of expression at which the DE-AA mutant could provide the essential C11 function without lethal toxicity remains to be determined."

We note here that our data did not address this possibility because of multiple confounding issues. The first is that the natural course of the plasmid shuffle approach which led to the C11 NTD-L as sufficient for viability relied on over-expression of the mutants, which also revealed and confirmed that the C11_DE-AA is toxic/lethal when over-expressed. Original description of C11_DE-AA shuffle by Chedin et al. (1998) used similar over-expression came to this conclusion. I believe that toxicity of C11_DE-AA would be decreased at lower expression levels to a point where it may indeed rescue essentiality. Attempts at such experiments are described in the Chedin et al paragraph that begins with "This model implies" whose 'data not shown' is cited to conclude that 1) C11_DE-AA mutant is toxic in a dose-dependent manner and 2) the DE-AA mutation is lethal when not over-expressed. The latter was based on plasmid-mediated expression under promoter-repressive conditions and observed inability of the yeast to lose the WT *RPC11* allele from the shuffle plasmid. The experiments required complex conditions involving nutrient-controlled over-expression and promoter repression. Yet there was no assurance that mutant C11 was expressed at all under the repressive conditions or at levels comparable to WT.

Therefore, what is needed is a plasmid shuffle or other type titration of the C11_DE-AA mutant in a viability rescue assay. Again, I suspect that C11-(DE-AA) may rescue at low level expression before becoming toxic and lethal as levels increase. It would be a nice experiment, although perhaps not trivial to undertake even with new technology ~25 years after the first attempts. Nonetheless, we have not tried it in part because we wanted to do it as part of a larger approach, together with NTD-L and other constructs, mostly to ask if over-expression of NTD-L is required for its ability to rescue viability. Although such a larger experiment would have been ideal, it became too large a goal as the COVID pandemic severely restricted laboratory access at the NIH and we found ourselves desperately trying to gather and tie-up loose ends rather than expand to new experimental projects. Sixteen months later things are different, including that no one who was working on this project is presently part of the lab (i.e., all coauthors of this manuscript except the corresponding author). These are important questions that we would like to address in the future.

8.) For the cleavage assays, no C37/C53 was used, which is puzzling at first because it also attributes to the binding interface with the C11 NTD. If I understood correctly, the authors later on in that paragraph argue that C37/C53 was omitted intentionally to assess direct binding of C11 to the Pol III core. I suggest mentioning that fact earlier. In addition, the manuscript might benefit from repeating that experiment with and without C37/C53 to compare activities of WT and mutant C11 constructs in absence and presence of C37/C53.

•Author response: We thank the Reviewer for the suggestion to mention earlier when describing these results that C37/53 was intentionally omitted from the cleavage assay. Because this approach might be considered similar to prior work in which purified Pol II enzyme was supplemented with individual recombinant domains of TFIS, we added a citation to that work (ref 55, Guo and Price, DH), “Our goal was to determine which of the C11 constructs could exhibit basic cleavage activity, somewhat similar to work in which purified Pol II ECs on tailed templates were supplemented with recombinant domains of TFIS⁵⁵. Therefore, C37/53 was not included in Fig. 2d which demonstrated active site cleavage by C11, CTD-L and CTD by producing cleavage products (CP) in the presence of MgCl₂, whereas C11-(DE-AA), NTD and NTD-L were inactive” (p. 7).

With regard to the additional point, we agree that it would be good to results of a cleavage assay with and without C37/C53 for the C11 mutants. This would extend the results by adding qualitative patterns of cleavage bands, as well as the dose-dependent effects on cleavage activities as in the Fig. 2e graph. However, doing this using the artificial backtracked EC (BEC) substrate might reveal only partial insight as compared to elongation assays in which Pol III may arrest upon encountering a terminator. It isn’t clear that reconstituting C37/53 onto an artificial BEC and then testing the various C11 constructs would lead to the same meaningful results as if a reconstituted Pol III EC would come to arrest upon encountering a terminator. Therefore, we believe that such experiments should be done in parallel to examine these possibilities but are presently beyond the scope of this manuscript, including in part because of disruption due to the extended nature of the COVID pandemic and that the expertise needed to do these experiments with training is no longer in the lab. We believe that addition of the comment referring to our approach without C37/53 is somewhat similar to what was done for TFIS (using what would appear to be Pol II core) is appropriate context. We thank the Reviewer for these comments.

9.) Fig. S5 showing the structure of hPol III in different C11 CTD positions should be revised as it is extremely difficult to recognize the features of importance (such as the C11 linker).

•Author response: We appreciate this comment. We realize in retrospect that the figure which superimposed models of both EC conformations of RPC10, “inside” and “outside” the funnel was chaotic and compromised views of each by trying to show both. We prepared a new Supplementary Fig. 7 in which only “inside the funnel” is used to aid the topics in the Discussion. A series of views were prepared as panels for Supplementary Fig. 7 to best try and lead the reader through the Discussion points. Supplementary Fig. 6 was added because it provides an extended version of the multiple sequence alignment of the funnel region which highlights amino acids in the Supplementary Fig. 7e/f structures that are not present in the shorter aligned regions in the cited work³⁴.

Minor points:

1.) Fig. 3C is missing the label “C11” below “C37/53” and the 1st “-“ in the row.

•Author response: We thank the Reviewer for noting this. It has been fixed.

2.) Fig. 8A is not referenced.

•Author response: We thank the Reviewer for noting and commenting on this. We now clearly refer to Figure 8a, and also more distinctly refer to plasmid shuffle per se, the topic of Fig. 8a.

3.) In the introduction, the structure of hPol III having the C11 CTD inserted into the funnel domain is mentioned twice in two separate paragraphs.

•Author response: We thank the Reviewer for noting and commenting on this. This duplication reflected a suboptimal passage which we revised to be more focused, articulate and clear.

4.) In the introduction, the essential C11 function are only briefly mentioned at the end and should be explained in greater detail (including primary citations).

•Author response: We appreciate this comment. We believe that it was very helpful toward an improved manuscript presented with better context. The revised Introduction includes the detailed information on C11 as suggested and is longer, but some of it had been in opening paragraphs of Results and Discussion sections which was moved forward.

5.) Figure 5: label (A) is shown in two panels. The second (A) in the figure on the bottom left should be changed to (B). The (B) in the right panel should be deleted.

•Author response: We thank the Reviewer for noting our labelling error. It has been corrected.

6.) On p13, lane 6, the Fig. reference is missing the actual figure number. (line 6)

•**Author response:** We thank the Reviewer for noting our typographical error. It has been fixed.

7.) At the bottom of p14/beginning of p15, the authors refer to northern blotting experiments but don't refer to the respective figure (I assume Fig. 8F?).

•**Author response:** We thank the Reviewer for noting our typographical error of omission. We have fixed this in the revised manuscript.

Reviewer #2 (Remarks to the Author):

General Comments

This manuscript from Mishra and colleagues evaluates domain-specific functions of the small, but important C11 subunit of RNA polymerase 3. Using a blend of biochemical and genetic assays, this manuscript assigns unique functions to the N-terminal domain, the linker, and the C-terminal domain of C11.

RNA polymerase 3 is unique in its ability to reinitiate transcription immediately after termination. There is a robust history/literature focused on the roles of individual subunits in this activity as well as the unique termination mechanism presented by the enzyme.

Here, Mishra and colleagues show that 3' RNA cleavage is in fact not essential for RNA polymerase 3, but that termination and recycling are. This conclusion refines previous interpretation of genetic data.

In general, the story is interesting and precise. The findings build on a long history of work with this enzyme, and the lab is clearly expert in the details. Related to that perceived strength, however, it is hard to conclude that this manuscript will appeal to a broad readership like that of Nature Communications. As this study refines many previous interpretations (based on genetics and biochemistry), it seems more appropriate for a specialized journal.

•**Author response:** We thank this Reviewer for careful reading of our paper, the expertise in the unique aspects of Pol III and the advances made in this study. With regard to appropriateness for broad readership, we added more details related to Biology and provided comments that may provide new insights into that of the other Pols.

Specific Comments

1. Many (most) of the data are representative and very little quantification or evaluation of reproducibility is displayed or described. Figure 4 is of particular concern since the slight lag in product is interpreted as an effect on binding. Although this interpretation is reasonable, it is not justified without consideration of variability in the results. In short, error analysis and reproducibility is not clear throughout.

•**Author response:** With regard to the particular concern about an unjustified interpretation from data in Figure 4 we deleted the sentence referred to "The shape of the time course curves of the less productive reactions (Fig. 4c) is consistent with relatively slow engagement/recruitment of the initiation complexes by their Pol III."

We appreciate concern regarding lack of display or description of reproducibility. In describing Fig. 4c we added a sentence referring to the reproducible nature of data (p. 9, bottom).

To address general concerns of this Reviewer regarding "very little quantification" and "error analysis," we added multiple new figures/figure elements to the revised manuscript: i) regression analysis of the quantified data in Fig. 4c are displayed in expanded form with trendlines with R² values as new Supplementary Fig. 3, ii) *p*-values were added to the Fig. 3c graph, and iii) regression analysis of quantified data in Fig. 2e are displayed in expanded form with trendlines with R² values as new Supplementary Fig. 1b.

We believe that these revisions strengthen the manuscript and thank the Reviewer for this.

2. The timing of experiments should be more clearly explained in the figure and/or legend. The methods section is pretty clear, but some of the assays are difficult to interpret without clear description of timing.

•**Author response:** We thank the Reviewer for this helpful comment. We added schematic timeline descriptions of the experimental design to two of the figures, as part of panels A of Figs 3 and 4. We believe that these help to provide more clear explanation, in the figures as suggested.

3. Where is CTD alone in figure 2C?

•**Author response:** We added data obtained with construct CTD as lanes 13 & 14, and appropriately noted it in the legend.

4. Overall the results section is difficult to follow. It is written such that an expert in RNA polymerase 3 might understand and evaluate impact, but it is very difficult for someone outside of the field.

•**Author response:** We reviewed and revised the results accordingly to try to accommodate readers outside the Pol III field. Our goal was that it will be more inclusive, and hope that we achieved this.

5. The Discussion section is not as well refined grammatically, and should be edited.

•**Author response:** We have edited the revised Discussion accordingly. We thank the Reviewer for helpful comments. Again, we hope that we achieved high impact grammatical communication.

REVIEWERS' COMMENTS

Reviewer #1 (Remarks to the Author):

Overall, the authors have well addressed the different points raised by the referees. The manuscript has considerably improved and can now be accepted in Nature Communications.

Remaining issues that the authors should still address:

- Line 576: an adverb is missing in this sentence.

- Lines 575-584: This last paragraph discusses the difference between Pol III-holo and reconstituted Pol III-core+C11+C37/53. The paragraph appears to be tacked onto the end of the discussion without being well connected to the previous paragraphs. It is also rather speculative. I therefore suggest removing this paragraph.

- Fig. S7 could still be further improved by using a light grey or white background in all panels. It would be also clearer for non-expert readers if only the relevant elements are depicted in bright colors, whereas the rest is depicted in moderate colors. There are also a number of typos and formatting errors in the legend of Fig. S7.

Reviewer #2 (Remarks to the Author):

The authors have made substantial revisions to the manuscript in response to the comments raised by both previous reviewers.

All of these changes are appreciated, but they do not mitigate the concern that this study is a bit specialized for this journal, in the eyes of this reviewer.

Nevertheless, the revised manuscript is improved and addresses most of the previous concerns raised.

Point-by-point responses to the reviewers' comments, reproduced verbatim.
Reviewer comments verbatim in black font, author response in blue font

Reviewer #1 (Remarks to the Author):

Overall, the authors have well addressed the different points raised by the referees. The manuscript has considerably improved and can now be accepted in Nature Communications.

•Author response: We appreciate this reviewer's comments above that summarize that the points raised by the referees have been well addressed in our revised manuscript.

Remaining issues that the authors should still address:

- Line 576: an adverb is missing in this sentence.

•Author response: We thank the reviewer for noting our grammatical error. The sentence was part of the paragraph that was removed as suggested in the next comment.

- Lines 575-584: This last paragraph discusses the difference between Pol III-holo and reconstituted Pol III-core+C11+C37/53. The paragraph appears to be tacked onto the end of the discussion without being well connected to the previous paragraphs. It is also rather speculative. I therefore suggest removing this paragraph.

•Author response: We appreciate the reviewer's perspective and are thankful for the suggestion to remove the paragraph. After reviewing the Discussion, we agree. We removed what had been the last the paragraph of the Discussion.

- Fig. S7 could still be further improved by using a light grey or white background in all panels. It would be also clearer for non-expert readers if only the relevant elements are depicted in bright colors, whereas the rest is depicted in moderate colors. There are also a number of typos and formatting errors in the legend of Fig. S7.

•Author response: We thank the reviewer for the helpful suggestion that we agree improved the manuscript and will make it more clear for non-expert and all readers. We have left panel a as it was, with light grey background, and changed all other panels to white. In addition we decolorized (made white) all elements except the relevant ones in all other panels, b-f. We also thank this reviewer for careful reading the legends and noting the typos and other errors. These have been carefully reviewed, cross referenced including with the related Supp. Fig S6 and corrected. A small amount of text in the last section of the Discussion in which Supp S7 is introduced has been modified accordingly in the revised manuscript.

We would like to thank the reviewer for the many helpful comments throughout.

Reviewer #2 (Remarks to the Author):

The authors have made substantial revisions to the manuscript in response to the comments raised by both previous reviewers.

•Author response: We thank this reviewer for noting that we made substantial revisions in response to the previous reviewers' comments.

All of these changes are appreciated, but they do not mitigate the concern that this study is a bit specialized for this journal, in the eyes of this reviewer.

•Author response: We don't share the judgement that because the reviewer believes the study will not "appeal to a broad readership like that of Nature Communications" in part because it "refines many previous interpretations," and is therefore too specialized for this journal (quotes are from this reviewer's previous comments).

This is an opportunity for me to commend the Editors of *Nature Communications* for broadening the sights for their readership by considering this manuscript. In the eyes of this author, expanding on/adding to the mechanisms of RNA polymerase recycling would fit the broad category of transcription. I probably don't have to point out to the reviewer that not all of Pol II complexes are involved in transcribing highly unique mRNA genes that are principally regulated more at the assembly of the PIC rather than at the reinitiation stage. Here, I take the opportunity to remind the Editors that in its heyday (1980s-1990s), Pol III was *the* model for basic mechanisms of eukaryotic transcription. I believe that the results in this manuscript are very timely and will be very interesting to a broad readership in part

because its contents involve the so-called transcription initiation factor-like subunits C53/37 which are homologs of TFIIF subunits in the Pol II system, and the TFIIS homologs (C11/RPC10), and because it involves aspects of elongation control, a highly regulated phase of transcription in the Pol II system.

Nevertheless, the revised manuscript is improved and addresses most of the previous concerns raised.

•Author response: We thank this reviewer for noting our comments addressing the concerns raised and the positive aspects of our revised manuscript.